# TpopT: Efficient Trainable Template Optimization on Low-Dimensional Manifolds

## Abstract

In scientific and engineering scenarios, a recurring task is the detection of low-dimensional families of signals or patterns. A classic family of approaches, exemplified by template matching, aims to cover the search space with a dense template bank. While simple and highly interpretable, it suffers from poor computational efficiency due to unfavorable scaling in the signal space dimensionality. In this work, we study TpopT (TemPlate OPTimization) as an alternative scalable framework for detecting low-dimensional families of signals which maintains high interpretability. We provide a theoretical analysis of the convergence of Riemannian gradient descent for TpopT, and prove that it has a superior dimension scaling to covering. We also propose a practical TpopT framework for nonparametric signal sets, which incorporates techniques of embedding and kernel interpolation, and is further configurable into a trainable network architecture by unrolled optimization. The proposed trainable TpopT exhibits significantly improved efficiency-accuracy tradeoffs for gravitational wave detection, where matched filtering is currently a method of choice. We further illustrate the general applicability of this approach with experiments on handwritten digit data.

## 1 Introduction

*Low-dimensional structure* is ubiquitous in data arising from physical systems: these systems often involve relatively few intrinsic degrees of freedom, leading to low-rank Ji et al. (2010); Gibson et al. (2022), sparse Quan et al. (2015), or manifold structure Mokhtarian & Abbasi (2002); Brown et al. (2004); Lunga et al. (2013). In this paper, we study the fundamental problem of detecting and estimating signals which belong to a low-dimensional manifold, from noisy observations Wakin et al. (2005); Wakin (2007); Baraniuk & Wakin (2009).

Perhaps the most classical and intuitive approach to detecting families of signals is *matched filtering* (MF), which constructs a bank of templates, and compares them individually with the observation. Due to its simplicity and interpretability, MF remains the core method of choice in the gravitational wave detection of the scientific collaborations LIGO Abramovici et al. (1992); Abbott et al. (2015), Virgo Acernese et al. (2015) and KARGA Akutsu et al. (2021), where massive template banks are constructed to search for traces of gravitational waves produced by pairs of merging black holes in space Owen & Sathyaprakash (1999); Abbott et al. (2016; 2017); Yan et al. (2022b). Emerging advances on template placement (Roy et al., 2017; 2019) and optimization (Weerathunga & Mohanty, 2017; Pal & Nayak, 2023; Dal Canton et al., 2021) provide promising ideas of growth. The conceptual idea of large template banks for detection is also widely present in other scenarios such as neuroscience Shi et al. (2010), geophysics Caffagni et al. (2016); Rousset et al. (2017), image pose recognition Picos et al. (2016), radar signal processing Pardhu et al. (2014); Johnson (2009), and aerospace engineering Murphy et al. (2017). In the meantime, many modern learning architectures employ similar ideas of matching inputs with template banks, such as transformation-invariant neural networks which create a large number of templates by applying transformations to a smaller family of filters Sohn & Lee (2012); Kanazawa et al. (2014); Zhou et al. (2017).

One major limitation of this approach is its unfavorable scaling with respect to the signal manifold dimension. For gravitational wave detection, this leads to massive template banks in deployment, and presents a fundamental barrier to searching broader and higher dimensional signal manifolds. For

transformation-invariant neural networks, the dimension scaling limits their applications to relatively low-dimensional transformation groups such as rotations.

This paper is motivated by a simple observation: instead of using sample templates to cover the search space, we can search for a best-matching template via optimization over the search space with higher efficiency. In other words, while MF searches for the best-matching template by enumeration, a first-order optimization method can leverage the geometric properties of the signal set, and avoid the majority of unnecessary templates. We refer to this approach as template optimization (TpopT).

In many practical scenarios, we lack an analytical characterization of the signal manifold. We propose a nonparametric extension of TpopT, based on signal embedding and kernel interpolation, which retains the test-time efficiency of TpopT.[1] The components of this method can be trained on sample data, reducing the need for parameter tuning and improving the performance in Gaussian noise. Our training approach draws inspiration from unrolled optimization Chen et al. (2022), which treats the iterations of an optimization method as layers of a neural network. This approach has been widely used for estimating low-dimensional (sparse) signals Liu & Chen (2019); Xin et al. (2016) with promising results on a range of applications Monga et al. (2021); Diamond et al. (2017); Liang et al. (2019); Buchanan et al. (2022). The main contributions of this paper are as follows:

- Propose trainable TpopT as an efficient approach to detecting and estimating signals from low-dimensional families, with nonparametric extensions when an analytical data model is unavailable.
- Prove that Riemannian gradient descent for TpopT is exponentially more efficient than MF.
- Demonstrate significantly improved complexity-accuracy tradeoffs for gravitational wave detection, where MF is currently a method of choice.

## 2 PROBLEM FORMULATION AND METHODS

In this section, we describe the problem of detecting and recovering signals from a low-dimensional family, and provide a high-level overview of two approaches — matched filtering and template optimization (TpopT). The problem setup is simple: assume the signals of interest form a $d$-dimensional manifold $S \subset \mathbb{R}^D$, where $d \ll D$, and that they are normalized such that $S \subset \mathbb{S}^{D-1}$. For a given observation $\boldsymbol{x} \in \mathbb{R}^D$, we want to determine whether $\boldsymbol{x}$ consists of a noisy copy of some signal of interest, and recover the signal if it exists. More formally, we model the observation and label as:

$$\boldsymbol{x} = \begin{cases} a\,\boldsymbol{s}_\natural + \boldsymbol{z} & \text{if } y = 1 \\ \boldsymbol{z}, & \text{if } y = 0 \end{cases}.$$ (1)

where $a \in \mathbb{R}_+$ is the signal amplitude, $\boldsymbol{s}_\natural \in S$ is the ground truth signal, and $\boldsymbol{z} \sim \mathcal{N}(\boldsymbol{0}, \sigma^2 \boldsymbol{I})$. Our goal is to solve this detection and estimation problem with simultaneously high statistical accuracy and computational efficiency.

**Matched Filtering.** A natural decision statistic for this detection problem is $\max_{\boldsymbol{s} \in S} \langle \boldsymbol{s}, \boldsymbol{x} \rangle$, i.e.

$$\hat{y}(\boldsymbol{x}) = 1 \iff \max_{\boldsymbol{s} \in S} \langle \boldsymbol{s}, \boldsymbol{x} \rangle \geq \tau$$ (2)

where $\tau$ is some threshold, and the recovered signal can be obtained as $\arg\max_{\boldsymbol{s} \in S} \langle \boldsymbol{s}, \boldsymbol{x} \rangle$. [2] *Matched filtering*, or template matching, approximates the above decision statistic with the maximum over a finite bank of templates $\boldsymbol{s}_1, \ldots, \boldsymbol{s}_{n_{\text{templates}}}$:

$$\hat{y}_{\text{MF}}(\boldsymbol{x}) = 1 \iff \max_{i=1,\ldots,n_{\text{templates}}} \langle \boldsymbol{s}_i, \boldsymbol{x} \rangle \geq \tau.$$ (3)

The template $\boldsymbol{s}_i$ contributing to the highest correlation is thus the recovered signal. This matched filtering method is a fundamental technique in signal detection (simultaneously obtaining the estimated signals), playing an especially significant role in scientific applications Owen & Sathyaprakash (1999); Rousset et al. (2017); Shi et al. (2010).

---

[1]In contrast to conventional manifold learning, where the goal is to learn a representation of the data manifold Bengio & Monperrus (2004); Culpepper & Olshausen (2009); Rifai et al. (2011); Park et al. (2015); Kumar et al. (2017), our goal is to learn an *optimization algorithm* on the signal manifold.

[2]This statistic is optimal for detecting a single signal $\boldsymbol{s}$ in iid Gaussian noise; this is the classical motivation for matched filtering Helstrom (2013). For detecting a family of signals $\boldsymbol{s} \in S$, it is no longer statistically optimal Yan et al. (2022a). However, it remains appealing due to its simplicity.

If the template bank densely covers $S$, (3) will accurately approximate (2). However, dense covering is inefficient — the number $n$ of templates required to cover $S$ up to some target radius $r$ grows as $n \propto 1/r^d$, making this approach impractical for all but the smallest $d$.[3]

**Template Optimization.** Rather than densely covering the signal space, *template optimization* (TpopT) searches for a best matching template $\hat{s}$, by numerically solving

$$\hat{s}(\boldsymbol{x}) = \arg\min_{\boldsymbol{s} \in S} f(\boldsymbol{s}) \equiv -\langle \boldsymbol{s}, \boldsymbol{x} \rangle. \tag{4}$$

The decision statistic is then $\hat{y}_{\text{TpopT}}(\boldsymbol{x}) = 1 \iff \langle \hat{s}(\boldsymbol{x}), \boldsymbol{x} \rangle \geq \tau$. Since the domain of optimization $S$ is a Riemannian manifold, in principle, the optimization problem (4) can be solved by the Riemannian gradient iteration Boumal (2023)

$$\boldsymbol{s}^{k+1} = \exp_{\boldsymbol{s}^k}\left(-\alpha_k \operatorname{grad}[f](\boldsymbol{s}^k)\right). \tag{5}$$

Here, $k$ is the iteration index, $\exp_{\boldsymbol{s}}(\boldsymbol{v})$ is the exponential map at point $\boldsymbol{s}$, $\operatorname{grad}[f](\boldsymbol{s})$ is the Riemannian gradient[4] of the objective $f$ at point $\boldsymbol{s}$, and $\alpha_k$ is the step size.

Alternatively, if the signal manifold $S$ admits a global parameterization $\boldsymbol{s} = \boldsymbol{s}(\boldsymbol{\xi})$, we can optimize over the parameters $\boldsymbol{\xi}$, solving $\hat{\boldsymbol{\xi}}(\boldsymbol{x}) = \arg\min_{\boldsymbol{\xi}} -\langle \boldsymbol{s}(\boldsymbol{\xi}), \boldsymbol{x} \rangle$ using the (Euclidean) gradient method:

$$\boldsymbol{\xi}^{k+1} = \boldsymbol{\xi}^k + \alpha_k \cdot \left(\nabla \boldsymbol{s}(\boldsymbol{\xi}^k)\right)^{\mathrm{T}} \boldsymbol{x}, \tag{6}$$

where $\nabla \boldsymbol{s}(\boldsymbol{\xi}^k) \in \mathbb{R}^{D \times d}$ is the Jacobian matrix of $\boldsymbol{s}(\boldsymbol{\xi})$ at point $\boldsymbol{\xi}^k$. Finally, the estimated signal $\hat{\boldsymbol{s}}(\boldsymbol{x}) = \boldsymbol{s}(\hat{\boldsymbol{\xi}}(\boldsymbol{x}))$ and decision statistic $\hat{y}_{\text{TpopT}}$ can be obtained from the estimated parameters $\hat{\boldsymbol{\xi}}$.

Of course, the optimization problem (4) is in general nonconvex, and methods (5)-(6) only converge to global optima when they are initialized sufficiently close to the solution of (4). We can guarantee global optimality by employing multiple initializations $\boldsymbol{s}_1^0, \ldots, \boldsymbol{s}_{n_{\text{init}}}^0$, which cover the manifold $S$ at some radius $\Delta$ where at least one initialization is guaranteed to produce a global optimizer.

In the next section, we will corroborate these intuitions with rigorous analysis. In subsequent sections, we will further develop more practical counterparts to (5)-(6) which (i) do not require an analytical representation of the signal manifold $S$ [Section 4], and (ii) can be trained on sample data to improve statistical performance [Section 5].

## 3 Theory: Efficiency Gains over Matched Filtering

The efficiency advantage of optimization comes from its ability to use gradient information to rapidly converge to $\hat{\boldsymbol{s}} \approx \boldsymbol{s}_\natural$, within a basin of initializations $\boldsymbol{s}^0$ satisfying $d(\boldsymbol{s}^0, \boldsymbol{s}_\natural) \leq \Delta$: the larger the basin, the fewer initializations are needed to guarantee global optimality. The basin size $\Delta$ in turn depends on the geometry of the signal set $S$, through its *curvature*. Figure 1 illustrates the key intuition: if the curvature is small, there exists a relatively large region in which the gradient of the objective function points towards the global optimizer $\boldsymbol{s}^\star$. On the other hand, if the signal manifold is very curvy, there may only exist a relatively small region in which the gradient points in the correct direction.

We can formalize this intuition through the curvature of geodesics on the manifold $S$. For a smooth curve $\gamma : [0, T] \to S \subset \mathbb{R}^n$, with unit speed parameterization $\boldsymbol{\gamma}(t)$, $t \in [0, T]$, the maximum curvature is

$$\kappa(\boldsymbol{\gamma}) = \sup_{t \in T} \|\ddot{\boldsymbol{\gamma}}(t)\|_2. \tag{7}$$

Geometrically, $\kappa^{-1}$ is the minimum, over all points $\boldsymbol{\gamma}(t)$, of the radius of the osculating circle whose velocity and acceleration match those of $\boldsymbol{\gamma}$ at $t$. We extend this definition to $S$, a Riemannian submanifold of $\mathbb{R}^n$, by taking $\kappa$ to be the maximum curvature of any geodesic on $S$:

$$\kappa(S) = \sup_{\boldsymbol{\gamma} \subset S \,:\, \text{unit-speed geodesic}} \kappa(\boldsymbol{\gamma}). \tag{8}$$

---

[3]This inefficiency has motivated significant efforts in applied communities to optimize the placement of the templates $\boldsymbol{s}_i$, maximizing the statistical performance for a given fixed $n_{\text{templates}}$ Owen & Sathyaprakash (1999). It is also possible to learn these templates from data, leveraging connections to neural networks Yan et al. (2022a). Nevertheless, the curse of dimensionality remains in force.

[4]The Riemannian gradient is the projection of the Euclidean gradient $\nabla_{\boldsymbol{s}} f$ onto the tangent space $T_{\boldsymbol{s}} S$.

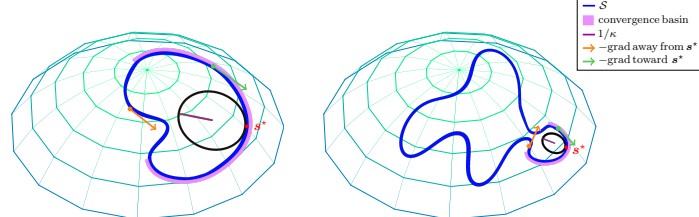

Figure 1: Relationship between curvature and convergence basins of gradient descent. Gradient descent has larger convergence basins under lower curvature (larger radius of osculating circle). Points within the convergence basin have gradient descent direction pointing "toward" $s^\star$, while points outside the basin may have gradient descent pointing "away from" $s^\star$.

We call this quantity the *extrinsic geodesic curvature* of $S$.[5] Our main theoretical result shows that, as suggested by Figure 1 there is a $\Delta = 1/\kappa$ neighborhood within which gradient descent rapidly converges to a close approximation of $s_\natural$:

**Theorem 1.** *Suppose the extrinsic geodesic curvature of $S$ is bounded by $\kappa$. Consider the Riemannian gradient method* (5), *with initialization satisfying $d(s^0, s_\natural) < 1/\kappa$, and step size $\tau = \frac{1}{64}$. Then when $\sigma \le c/(\kappa\sqrt{d})$, with high probability, we have for all $k$*

$$d(s^{k+1}, s_\natural) \le (1 - \epsilon)\, d(s^k, s_\natural) + C\sigma\sqrt{d}. \tag{9}$$

*Moreover, when $\sigma \le c/(\kappa\sqrt{D})$, with high probability, we have for all $k$*

$$d(s^k, s^\star) \le C(1 - \epsilon)^k \sqrt{f(s^0) - f(s^\star)}, \tag{10}$$

*where $s^\star$ is the unique minimizer of $f$ over $B(s_\natural, 1/\kappa)$. Here, $C, c, \epsilon$ are positive numerical constants.*

**Interpretation: Convergence to Optimal Statistical Precision** In (9), we show that under a relatively mild condition on the noise, gradient descent exhibits linear convergence to a $\sigma\sqrt{d}$-neighborhood of $s_\natural$. This accuracy is the best achievable up to constants: for small $\sigma$, with high probability any minimizer $s^\star$ satisfies $d(s^\star, s_\natural) > c\sigma\sqrt{d}$, and so the accuracy guaranteed by (9) is optimal up to constants. Also noteworthy is that both the accuracy and the required bound on the noise level $\sigma$ are *dictated solely by the intrinsic dimension $d$*. The restriction $\sigma \le c/(\kappa\sqrt{D})$ has a natural interpretation in terms of Figure 1 — at this scale, the noise "acts locally", ensuring that $s^\star$ is close enough to $s_\natural$ so that for any initialization in $B(s_\natural, \Delta)$, the gradient points toward $s^\star$. In (10) we also show that under a stronger condition on $\sigma$, gradient descent enjoys linear convergence for *all* iterations $k$.

**Implications on Complexity.** Here we compare the complexity required for MF and TpopT to achieve a target estimation accuracy $d(\hat{s}, s_\natural) \le r$. The complexity of MF is simply $N_r$, the covering number of $S$ with radius $r$. On the other hand, the complexity of TpopT is dictated by $n_{\text{init}} \times n_{\text{gradient-step}}$. We have $n_{\text{init}} = N_{1/\kappa}$ since TpopT requires initialization within radius $1/\kappa$ of $s_\natural$, and $n_{\text{gradient-step}} \propto \log 1/\kappa r$ because gradient descent enjoys a linear convergence rate. Note that the above argument applies when $C\sigma d^{1/2}/\epsilon \le r \le 1/\kappa$, where the upper bound on $r$ prescribes the regime where gradient descent is in action (otherwise TpopT and MF are identical), and the lower bound on $r$ reflects the statistical limitation due to noise. Since the covering number $N_{\text{radius}} \propto (1/\text{radius})^d$, the complexities of the two methods $T_{\text{MF}}$ and $T_{\text{TpopT}}$ are given by

$$T_{\text{MF}} \propto 1/r^d, \qquad T_{\text{TpopT}} \propto \kappa^d \log(1/\kappa r). \tag{11}$$

Combining this with the range of $r$, it follows that TpopT always has superior dimensional scaling than MF whenever the allowable estimation error $r$ is below $1/\kappa$ (and identical to MF above that). The advantage is more significant at lower noise and higher estimation accuracy.

---

[5]Notice that $\kappa(S)$ measures how $S$ curves *in the ambient space* $\mathbb{R}^n$; this is in contrast to traditional *intrinsic* curvature notions in Riemannian geometry, such as the sectional and Ricci curvatures. An extrinsic notion of curvature is relevant here because our objective function $f(s) = -\langle s, x \rangle$ is defined extrinsically. Intrinsic curvature also plays an important role in our arguments — in particular, in controlling the effect of noise.

**Proof Ideas.** The proof of Theorem 1 follows the intuition in Figure 1, by (i) considering a noiseless version of the problem and showing that in a $1/\kappa$ ball, the gradient points towards $s_\natural$, and (ii) controlling the effect of noise, by bounding the maximum component $T^{\max}$ of the noise $z$ along any tangent vector $v \in T_s S$ at any point $s \in B(s_\natural, \Delta)$. By carefully controlling $T^{\max}$, we are able to achieve rates driven by intrinsic dimension, not ambient dimension. Intuitively, this is because the collection of tangent vectors, i.e., the tangent bundle, has dimension $2d$. Our proof involves a discretization argument, which uses elements of Riemannian geometry (Toponogov's theorem on geodesic triangles, control of parallel transport via the second fundamental form Toponogov (2006); Lee (2018)). To show convergence of iterates (10), we show that in a $1/\kappa$ region, the objective $f$ enjoys Riemannian strong convexity and Lipschitz gradients Boumal (2023). Please see the supplementary material for complete proofs.

## 4 NONPARAMETRIC TPOPT VIA EMBEDDING AND KERNEL INTERPOLATION

The theoretical results in Section 3 rigorously quantify the advantages of TpopT in detecting and estimating signals from low-dimensional families. A straightforward application of TpopT requires a precise analytical characterization of the signal manifold. In this section, we develop more practical, nonparametric extension of TpopT, which is applicable in scenarios in which we *only* have examples $s_1, \ldots, s_N$ from $S$. This extension will maintain the test-time efficiency advantages of TpopT.

**Embedding.** We begin by embedding the example points $s_1, \ldots, s_N \in \mathbb{R}^n$ into a lower-dimensional space $\mathbb{R}^d$, producing data points $\xi_1, \ldots, \xi_N \in \mathbb{R}^d$. The mapping $\varphi$ should preserve pairwise distances and can be chosen in a variety of ways; Because the classical Multidimensional Scaling (MDS) setup on Euclidean distances is equivalent to Principal Component Analysis (PCA), we simply use PCA in our experiments. Assuming that $\varphi$ is bijective over $S$, we can take $s = s(\xi)$ as an approximate parameterization of $S$, and develop an optimization method which, given an input $x$, searches for a parameter $\xi \in \mathbb{R}^d$ that minimizes $f(s(\xi)) = -\langle s(\xi), x \rangle$.

**Kernel Interpolated Jacobian Estimates.** In the nonparameteric setting, we only know the values of $s(\xi)$ at the finite point set $\xi_1, \ldots, \xi_N$, and we do not have any direct knowledge of the functional form of the mapping $s(\cdot)$ or its derivatives. To extend TpopT to this setting, we can estimate the Jacobian $\nabla s(\xi)$ at point $\xi_i$ by solving a weighted least squares problem

$$\widehat{\nabla s}(\xi_i) = \arg \min_{J \in \mathbb{R}^{D \times d}} \sum_{j=1}^{N} w_{j,i} \|s_j - s_i - J(\xi_j - \xi_i)\|_2^2, \tag{12}$$

where the weights $w_{j,i} = \Theta(\xi_i, \xi_j)$ are generated by an appropriately chosen kernel $\Theta$. The least squares problem (12) is solvable in closed form. In practice, we prefer compactly supported kernels, so that the sum in (12) involves only a small subset of the points $\xi_j$;[6] in experiment, we choose $\Theta$ to be a truncated radial basis function kernel $\Theta_{\lambda,\delta}(x_1, x_2) = \exp(-\lambda \|x_1 - x_2\|_2^2) \cdot \mathbb{1}_{\|x_1 - x_2\|_2 < \delta}$. When example points $s_i$ are sufficiently dense and the kernel $\Theta$ is sufficiently localized, $\widehat{\nabla s}(\xi)$ will accurately approximate the true Jacobian $\nabla s(\xi)$.

**Expanding the Basin of Attraction using Smoothing.** In actual applications such as computer vision and astronomy, the signal manifold $S$ often exhibits large curvature $\kappa$, leading to a small basin of attraction. One classical heuristic for increasing the basin size is to *smooth* the objective function $f$. We can incorporate smoothing by taking gradient steps with a kernel smoothed Jacobian,

$$\widetilde{\nabla s}(\xi_i) = Z^{-1} \sum_j w_{j,i} \widehat{\nabla s}(\xi_j), \tag{13}$$

where $w_{j,i} = \Theta_{\lambda_s, \delta_s}(\xi_i, \xi_j)$ and $Z = \sum_j w_{j,i}$. The gradient iteration becomes

$$\xi^{k+1} = \xi^k + \alpha_k \widetilde{\nabla s}(\xi^k)^\mathrm{T} x. \tag{14}$$

When the Jacobian estimate $\widehat{\nabla s}(\xi)$ accurately approximates $\nabla s(\xi)$, we have

$$\widetilde{\nabla s}(\xi_i)^\mathrm{T} x \approx Z^{-1} \sum_j w_{j,i} \nabla s(\xi_j)^\mathrm{T} x = \nabla \Big[ Z^{-1} \sum_j w_{j,i} f(s(\xi_j)) \Big]. \tag{15}$$

---

[6]In our experiments on gravitational wave astronomy, we introduce an additional quantization step, computing approximate Jacobians on a regular grid $\hat{\xi}_1, \ldots, \hat{\xi}_{N'}$ of points in the parameter space $\Xi$.

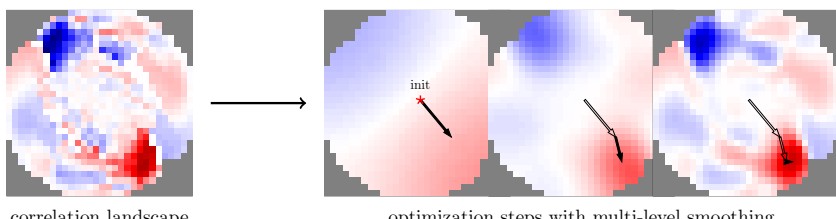

correlation landscape                    optimization steps with multi-level smoothing

Figure 2: Illustration of 2-dim signal embeddings and the parameter optimization procedure for gravitational wave signals.

i.e., $\widetilde{\nabla s}^{\mathrm{T}}$ is an approximate gradient for a smoothed version $\widetilde{f}$ of the objective $f$. Figure 2 illustrates smoothed optimization landscapes $\widetilde{f}$ for different levels of smoothing, i.e., different choices of $\lambda_s$. In general, the more smoothing is applied, the broader the basin of attraction. We employ a coarse-to-fine approach, which starts with a highly smoothed landscape (small $\lambda_s$) in the first iteration and decreases the level of smoothing from iteration to iteration — see Figure 2.

These observations are in line with theory: because our embedding approximately preserves Euclidean distances, $\|\boldsymbol{\xi}_i - \boldsymbol{\xi}_j\|_2 \approx \|\boldsymbol{s}_i - \boldsymbol{s}_j\|_2$, we have

$$\widetilde{f}(\boldsymbol{s}(\boldsymbol{\xi}_i)) = Z^{-1} \sum_j \Theta(\boldsymbol{\xi}_i, \boldsymbol{\xi}_j) \langle \boldsymbol{s}_j, \boldsymbol{x} \rangle \approx \langle Z^{-1} \sum_j \Theta(\boldsymbol{s}_i, \boldsymbol{s}_j) \boldsymbol{s}_j, \boldsymbol{x} \rangle, \tag{16}$$

i.e., applying kernel smoothing in the parameter space is nearly equivalent to applying kernel smoothing to the signal manifold $S$. This smoothing operation expands the basin of attraction $\Delta = 1/\kappa$, by reducing the manifold curvature $\kappa$. Empirically, we find that with appropriate smoothing often a single initialization suffices for convergence to global optimality, suggesting this as a potential key to breaking the curse of dimensionality.

## 5 TRAINING NONPARAMETRIC TPOPT

In the section above, we described nonparametric TpopT for finding the matching template by the iterative gradient solver (5). Note that this framework requires pre-computing the Jacobians $\nabla \boldsymbol{s}(\boldsymbol{\xi})$ and determining optimization hyperparameters, including the step sizes $\alpha_k$ and kernel width parameters $\lambda_k$ at each layer. In this section, we adapt TpopT into a trainable architecture, which essentially learns all the above quantities from data to further improve performance.

Recall the gradient descent iteration (14) in TpopT. Notice that if we define a collection of matrices $\boldsymbol{W}(\boldsymbol{\xi}_i, k) = \alpha_k \widetilde{\nabla s}(\boldsymbol{\xi}_i)^{\mathrm{T}} \in \mathbb{R}^{d \times D}$ indexed by $\boldsymbol{\xi}_i \in \{\boldsymbol{\xi}_1, \ldots, \boldsymbol{\xi}_N\}$ and $k \in \{1, \ldots, K\}$ where $K$ is the total number of iterations, then the iteration can be rewritten as

$$\boldsymbol{\xi}^{k+1} = C^{-1} \sum_{i=1}^{N} w_{k,i} \left( \boldsymbol{\xi}_i + \boldsymbol{W}(\boldsymbol{\xi}_i, k) \, \boldsymbol{x} \right), \tag{17}$$

where $w_{k,i} = \Theta_{\lambda_k, \delta_k}(\boldsymbol{\xi}^k, \boldsymbol{\xi}_i)$, and $C = \sum_i w_{k,i}$. Equation (17) can be interpreted as a kernel interpolated gradient step, where the $\boldsymbol{W}$ matrices summarize the Jacobian and step size information. Because $\Theta$ is compactly supported, this sum involves only a small subset of the sample points $\boldsymbol{\xi}_i$. Now, if we "unroll' the optimization by viewing each gradient descent iteration as one layer of a trainable network, we arrive at a trainable TpopT architecture, as illustrated in Figure 3. Here the trainable parameters in the network are the $\boldsymbol{W}(\boldsymbol{\xi}_i, k)$ matrices and the kernel width parameters $\lambda_k$.

Following our heuristic that the $\boldsymbol{W}(\boldsymbol{\xi}_i, k)$ matrices were originally the combination of Jacobian and step size, we can initialize these matrices as $\alpha_k \widetilde{\nabla s}(\boldsymbol{\xi}_i)^{\mathrm{T}}$. For the loss function during training, we use the square loss between the network output $\boldsymbol{\xi}^K(\boldsymbol{x})$ and the optimal quantization point $\boldsymbol{\xi}^*(\boldsymbol{x}) = \arg\max_{i=1,\ldots,N} \langle \boldsymbol{s}(\boldsymbol{\xi}_i), \boldsymbol{x} \rangle$, namely

$$L = \frac{1}{N_{\mathrm{train}}} \sum_{j=1}^{N_{\mathrm{train}}} \|\boldsymbol{\xi}^K(\boldsymbol{x}_j) - \boldsymbol{\xi}^*(\boldsymbol{x}_j)\|_2^2 \tag{18}$$

for a training set $\{\boldsymbol{x}_j\}_{j=1}^{N_{\mathrm{train}}}$ with positively-labeled data only. This loss function is well-aligned with the signal estimation task, and is also applicable to detection.

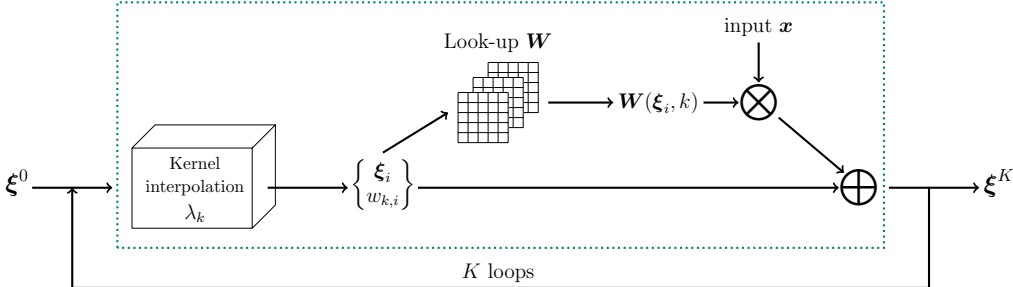

Figure 3: Architecture of trainable TpopT. The model takes $x$ as input and starts with a fixed initialization $\boldsymbol{\xi}^0$, and outputs $\boldsymbol{\xi}^K$ after going through $K$ layers. The trainable parameters are the collection of $\boldsymbol{W}(\boldsymbol{\xi}_i, k)$ matrices and kernel width parameters $\lambda_k$.

In summary, the trainable TpopT architecture consists of the following steps:

- Create embeddings $\boldsymbol{s}_i \mapsto \boldsymbol{\xi}_i$.
- Estimate Jacobians $\nabla \boldsymbol{s}(\boldsymbol{\xi})$ at points $\boldsymbol{\xi}_i$ by weighted least squares.
- Estimate smoothed Jacobians $\widetilde{\nabla \boldsymbol{s}}(\boldsymbol{\xi})$ at any $\boldsymbol{\xi}$ by kernel smoothing.
- Select a multi-level smoothing scheme.
- Train the model with unrolled optimization.

## 6 EXPERIMENTS

We apply the trainable TpopT to gravitational wave detection, where MF is the current method of choice, and show a significant improvement in efficiency-accuracy tradeoffs. We further demonstrate its wide applicability on low-dimensional data with experiments on handwritten digit data.

To compare the efficiency-accuracy tradeoffs of MF and TpopT models, we note that (i) for MF, the computation cost of the statistic $\max_{i=1,\ldots,n} \langle \boldsymbol{s}_i, \boldsymbol{x} \rangle$ is dominated by the cost of $n$ length $D$ inner products, requiring $nD$ multiplication operations. For TpopT, with $M$ parallel initializations, $K$ iterations of the gradient descent (17), $m$ neighbors in the truncated kernel, and a final evaluation of the statistic, we require $MD(Kdm + 1)$ multiplications; other operations including the kernel interpolation and look-up of pre-computed gradients have negligible test-time cost.

### 6.1 GRAVITATIONAL WAVE DETECTION

We aim to detect a family of gravitational wave signals in Gaussian noise. Each gravitational wave signal is a one-dimensional chirp-like signal – see Figure 4 (left).[7] Please refer to section F in the appendix for data generation details.

Based on their physical modeling, gravitational wave signals are equipped with a set of physical parameters, such as the masses and three-dimensional spins of the binary black holes that generate them, etc. While it is tempting to directly optimize on this native parameter space, unfortunately the optimization landscape on this space turns out to be rather unfavorable, as shown in Figure 4 (right). We see that the objective function has many spurious local optimizers and is poorly conditioned. Therefore, we still resort to signal embedding to create an alternative set of approximate "parameters" that are better suited for optimization.

For the signal embedding, we apply PCA with dimension 2 on a separate set of 30,000 noiseless waveforms drawn from the same distribution. Because the embedding dimension is relatively low, here we quantize the embedding parameter space with an evenly-spaced grid, with the range of each dimension evenly divided into 30 intervals. The value $\boldsymbol{\xi}^0$ at the initial layer of TpopT is fixed at the center of this quantization grid. Prior to training, we first determine the optimization hyperparameters

---

[7]The raw data of gravitational wave detection is a noisy one-dimensional time series, where gravitational wave signals can occur at arbitrary locations. We simplify the problem by considering input segments of fixed time duration.

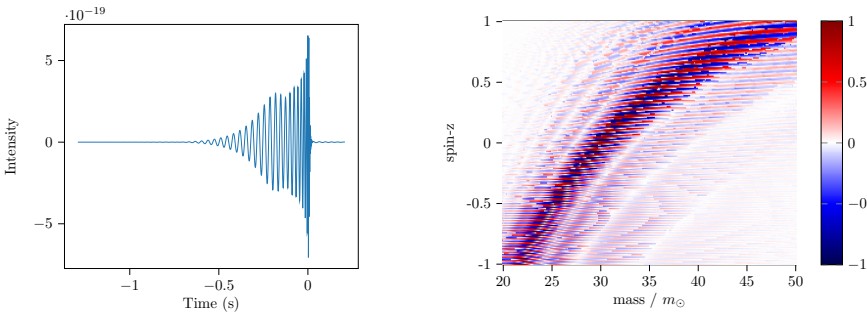

Figure 4: **Left:** Example of a gravitational wave signal. **Right:** Optimization landscape in the physical parameter space (mass-spin-$z$), shown as the heatmap of signal correlations.

(step sizes and smoothing levels) using a layer-wise greedy grid search, where we sequentially choose the step size and smoothing level at each layer as if it were the final layer. This greedy approach significantly reduces the cost of the search. From there, we use these optimization hyperparameters to initialize the trainable TpopT network, and train the parameters on the training set. We use the Adam Kingma & Ba (2014) optimizer with batch size 1000 and constant learning rate $10^{-2}$. Regarding the computational cost of TpopT, we have $M = 1, d = 2, m = 4$ during training and $m = 1$ during testing. The test time complexity of $K$-layer TpopT is $D(2K + 1)$.

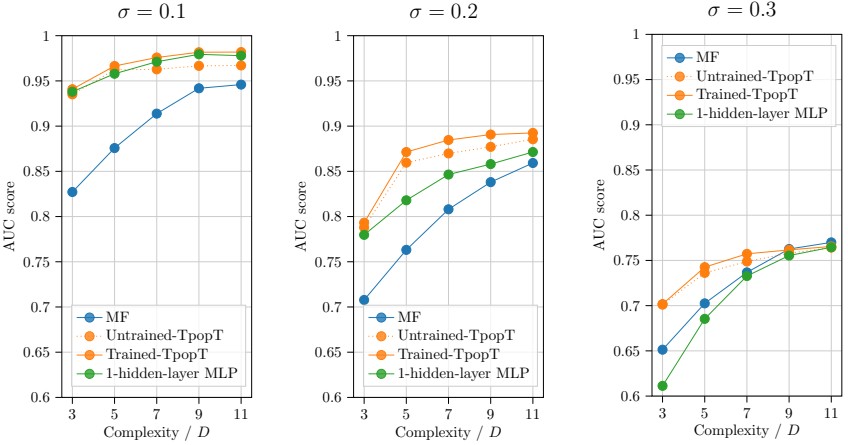

Figure 5: This figure compares the performance of four methods: (1) matched filtering (MF), (2) Template optimization (TpopT) without training, (3) TpopT with training, and (4) multi-layer perceptron (MLP) with one hidden layer. All methods are compared at three noise levels. We see that TpopT performs well in low to moderate noise, which matches theoretical results.

To evaluate the performance of matched filtering at any given complexity $m$, we randomly generate 1,000 independent sets of $m$ templates drawn from the above distribution, evaluate the ROC curves of each set of templates on the validation set, and select the set with the highest area-under-curve (AUC) score. This selected template bank is then compared with TpopT on the shared test set.

Figure 6.1 shows the comparison of efficiency-accuracy tradeoffs for this task between matched filtering and TpopT after training. We see that TpopT achieves significantly higher detection accuracy compared with MF at equal complexity. At low to moderate noise levels, Trained-TpopT performs the best, followed by MLP, and matched filtering performs the worst. As noise level increases, MLP's performance worsens most significantly, becoming the worst at $\sigma = 0.3$.

## 6.2 HANDWRITTEN DIGIT RECOGNITION

In this second experiment, we apply TpopT to the classic task of handwritten digit recognition using the MNIST Deng (2012) dataset, in particular detecting the digit 3 from all other digits. We apply random Euclidean transformations to all images, with translation uniformly between $\pm 0.1$ image size on both dimensions and rotation angle uniformly between $\pm 30°$.

Since the signal space here is nonparametric, we first create a 3-dimensional PCA embedding from the training set, and Figure 6 (left) shows a slice of the embedding projected onto the first two embedding dimensions. See supplementary for experiment details. Regarding the computational cost of TpopT, we have $M = 1$, $d = 3$, $m = 5$ during training and $m = 1$ during testing. Since the complexity is measured at test time, the complexity with $K$-layer TpopT is $D(3K + 1)$. Additional experimental details can be found in F.

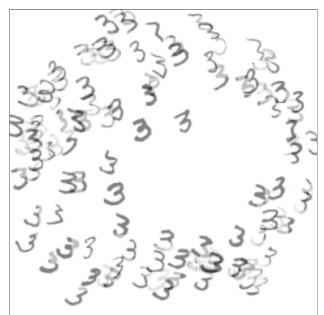 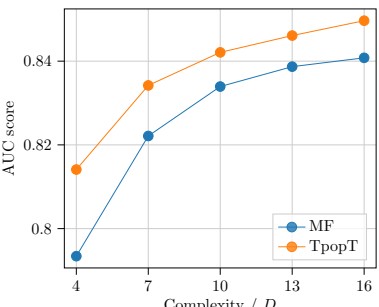

Figure 6: **Left:** A slice of the 3-d embeddings projected onto the first two dimensions. **Right:** Classification scores of MF and TpopT at different complexity levels, for handwritten digit recognition.

Matched filtering is also evaluated similarly as in the previous experiment. We first set aside a random subset of 500 images of digit 3 from the MNIST training set and construct the validation set from it. The remaining images are used to randomly generate 1,000 independent sets of transformed digits 3, and the best-performing set of templates on the validation set is selected as the MF template bank, and compared with TpopT on the shared test set. Figure 6 (right) shows the comparison of efficiency-accuracy tradeoffs between the two methods, and we see a consistently higher detection accuracy of trained TpopT over MF at equal complexities.

## 7 Discussion and Limitations

In this paper, we studied TpopT as an approach to efficient detection of low-dimensional signals. We provided a proof of convergence of Riemannian gradient descent on the signal manifold, and demonstrated its superior dimension scaling compared to MF. We also proposed the trainable TpopT architecture that can handle general nonparametric families of signals. Experimental results show that trained TpopT achieves significantly improved efficiency-accuracy tradeoffs than MF, especially in the gravitational wave detection task where MF is the method of choice.

The principal limitation of nonparametric TpopT is its storage complexity: it represents the manifold using a dense collection of points and Jacobians, with cost exponential in intrinsic dimension $d$. At the same time, we note that the same exponential storage complexity is encountered by matched filtering with a pre-designed template bank. In some sense, this exponential resource requirement reflects an intrinsic constraint of the signal detection problem, unless more structures within the signal space can be exploited. Both TpopT and its nonparametric extension achieve exponential improvements in test-time efficiency compared to MF; nevertheless, our theoretical results retain an exponential dependence on intrinsic dimension $d$, due to the need for multiple initializations. In experiments, the proposed smoothing allows convergence to global optimality from a single initialization. Our current theory does not fully explain this observation; this is an important direction for future work.

An advantage of MF not highlighted in this paper is its efficiency in handling noisy time series, using the fast Fourier transform. This enables MF to rapidly locate signals that occur at a-priori unknown spatial/temporal locations. Developing a convolutional version of TpopT with similar advantages is another important direction.

Finally, our gravitational wave experiments use synthetic data with known ground truth, in order to corroborate the key messages of this paper. Future experiments that explore broader and more realistic setups will be an important empirical validation of the proposed method.

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

## A    APPENDIX

## B    OVERVIEW

In the appendices, we will prove Theorem 1 from the main paper.

For the rest of the supplementary materials, Section C proves result (10) under the stricter constraint on the noise level $\sigma$, Section D bounds the effect of noise on the tangent bundle, and Section E uses this bound to prove result (9) under the looser constraint on the noise level.

## C    PROOF OF RESULT (10)

In this section, we state and prove one of the two parts of our main claims about gradient descent:

> **Theorem 2.** *Let $S$ be a complete manifold. Suppose the extrinsic geodesic curvature of $S$ is bounded by $\kappa$. Consider the Riemannian gradient method, with initialization satisfying $d(s^0, s_\natural) < \Delta = 1/\kappa$, and step size $\tau = \frac{1}{64}$. Then when $\sigma \leq 1/(60\kappa\sqrt{D})$, with probability at least $1 - e^{-D/2}$, we have for all $k$*
>
> $$d(s^k, s^\star) \leq \left(1 - \epsilon\right)^k d(s^0, s^\star), \tag{19}$$
>
> *where $s^\star$ is the unique minimizer of $f$ over $B(s_\natural, 1/\kappa)$. Here, $c, \epsilon$ are positive numerical constants.*

*Proof.* Since the closed neighborhood $B(s_\natural, 1/\kappa)$ is a compact set and $f$ is continuous, there must exist a minimizer of $f$ on $B(s_\natural, 1/\kappa)$, which we denote as $s^\star$. We will show that with high probability $s^\star$ does not lie on the boundary $\partial B(s_\natural, 1/\kappa)$. It suffices to show that $\forall s \in \partial B(s_\natural, 1/\kappa)$ :

$$\left\langle -\operatorname{grad}[f](s), \frac{\log_s s_\natural}{\|\log_s s_\natural\|_2} \right\rangle > 0, \tag{20}$$

namely that the gradient descent direction points inward the neighborhood for all points on the boundary. Here $\log_s : S \to T_s S$ denotes the logarithmic map at point $s \in S$. To show this, we have

$$\left\langle -\operatorname{grad}[f](s), \frac{\log_s s_\natural}{\|\log_s s_\natural\|_2} \right\rangle = \left\langle P_{T_s S}[s_\natural + z], \frac{\log_s s_\natural}{\|\log_s s_\natural\|_2} \right\rangle$$

$$= \left\langle s_\natural + z, \frac{\log_s s_\natural}{\|\log_s s_\natural\|_2} \right\rangle$$

$$\geq \left\langle s_\natural, \frac{\log_s s_\natural}{\|\log_s s_\natural\|_2} \right\rangle - \|z\|_2, \tag{21}$$

where the operator $P_{T_s S}[\cdot]$ denotes projection onto the tangent space at $s$, and we used the fact that $\log_s s_\natural \in T_s S$. Let $\gamma$ be a unit-speed geodesic of $S$ with $\gamma(0) = s_\natural$ and $\gamma(\Delta) = s$, the existence of which is ensured by the completeness of $S$. Hence $\dot{\gamma}(\Delta) = -\frac{\log_s s_\natural}{\|\log_s s_\natural\|_2}$. Using Lemma 3, it follows that

$$\left\langle s_\natural, \frac{\log_s s_\natural}{\|\log_s s_\natural\|_2} \right\rangle = \langle \gamma(0), -\dot{\gamma}(\Delta) \rangle \geq \Delta - \frac{1}{6}\kappa^2\Delta^3 = \frac{5}{6\kappa}. \tag{22}$$

Throughout this proof, we will use the result from measure concentration that $\|z\|_2 \leq 2\sigma\sqrt{D}$ with probability at least $1 - e^{-D/2}$ **?**. Hence with high probability we have $\left\langle -\operatorname{grad}[f](s), \frac{\log_s s_\natural}{\|\log_s s_\natural\|_2} \right\rangle \geq \frac{5}{6\kappa} - \frac{1}{30\kappa} > 0$. Therefore, with high probability $s^\star$ lies in the interior of $B(s_\natural, 1/\kappa)$, and hence the gradient vanishes at $s^\star$, i.e. $\operatorname{grad}[f](s^\star) = 0$.

Suppose we are currently at the $k$-th iteration with iterate $s^k$. Define $s_t = \exp_{s^k}\left(-t\operatorname{grad}[f](s^k)\right)$ with variable $t \in [0, \tau]$, and the next iterate can be represented as $s^{k+1} = s_\tau$. The global definition

of the exponential map is ensured by the completeness of $S$. We have that

$$
\begin{aligned}
d(\boldsymbol{s}^{k+1}, \boldsymbol{s}^{\star}) - d(\boldsymbol{s}^{k}, \boldsymbol{s}^{\star}) &= \int_0^\tau \frac{d}{dr} d(\boldsymbol{s}_r, \boldsymbol{s}^{\star}) \Big|_t \, dt \\
&= \int_0^\tau \left\langle \frac{d}{dr} \boldsymbol{s}_r \Big|_t, \frac{-\log_{\boldsymbol{s}_t} \boldsymbol{s}^{\star}}{\|\log_{\boldsymbol{s}_t} \boldsymbol{s}^{\star}\|_2} \right\rangle \, dt \\
&= \int_0^\tau \left\langle \Pi_{\boldsymbol{s}_t, \boldsymbol{s}^k} \{-\operatorname{grad}[f](\boldsymbol{s}^k)\}, \frac{-\log_{\boldsymbol{s}_t} \boldsymbol{s}^{\star}}{\|\log_{\boldsymbol{s}_t} \boldsymbol{s}^{\star}\|_2} \right\rangle \, dt \\
&= \int_0^\tau \left\langle -\operatorname{grad}[f](\boldsymbol{s}_t), \frac{-\log_{\boldsymbol{s}_t} \boldsymbol{s}^{\star}}{\|\log_{\boldsymbol{s}_t} \boldsymbol{s}^{\star}\|_2} \right\rangle \, dt \\
&\quad + \int_0^\tau \left\langle \operatorname{grad}[f](\boldsymbol{s}_t) + \Pi_{\boldsymbol{s}_t, \boldsymbol{s}^k} \{-\operatorname{grad}[f](\boldsymbol{s}^k)\}, \frac{-\log_{\boldsymbol{s}_t} \boldsymbol{s}^{\star}}{\|\log_{\boldsymbol{s}_t} \boldsymbol{s}^{\star}\|_2} \right\rangle \, dt \\
&= \int_0^\tau \left\langle P_{T_{\boldsymbol{s}_t} S}[\boldsymbol{s}^{\star}], \frac{-\log_{\boldsymbol{s}_t} \boldsymbol{s}^{\star}}{\|\log_{\boldsymbol{s}_t} \boldsymbol{s}^{\star}\|_2} \right\rangle \, dt \\
&\quad + \int_0^\tau \left\langle P_{T_{\boldsymbol{s}_t} S}[\boldsymbol{x} - \boldsymbol{s}^{\star}], \frac{-\log_{\boldsymbol{s}_t} \boldsymbol{s}^{\star}}{\|\log_{\boldsymbol{s}_t} \boldsymbol{s}^{\star}\|_2} \right\rangle \, dt \\
&\quad + \int_0^\tau \left\langle \operatorname{grad}[f](\boldsymbol{s}_t) + \Pi_{\boldsymbol{s}_t, \boldsymbol{s}^k} \{-\operatorname{grad}[f](\boldsymbol{s}^k)\}, \frac{-\log_{\boldsymbol{s}_t} \boldsymbol{s}^{\star}}{\|\log_{\boldsymbol{s}_t} \boldsymbol{s}^{\star}\|_2} \right\rangle \, dt.
\end{aligned}
\tag{23}
$$

The third equation holds because the velocity at the new point $\boldsymbol{s}_t$ is the same velocity vector at $\boldsymbol{s}^k$ but transported along the curve since there is no acceleration along the curve. The last equation follows from the fact that $\operatorname{grad}[f](\boldsymbol{s}_t) = -P_{T_{\boldsymbol{s}_t} S}[\boldsymbol{x}]$. In the following, we will bound the three terms in (23) separately.

For convenience, write

$$
d(t) = d(\boldsymbol{s}_t, \boldsymbol{s}^{\star})
\tag{24}
$$

so that $d(0) = d(\boldsymbol{s}^k, \boldsymbol{s}^{\star})$ and $d(\tau) = d(\boldsymbol{s}^{k+1}, \boldsymbol{s}^{\star})$.

For the integrand of the first term in (23), let $\boldsymbol{\gamma}$ be a unit-speed geodesic between $\boldsymbol{s}_t$ and $\boldsymbol{s}^{\star}$, where $\boldsymbol{\gamma}(0) = \boldsymbol{s}^{\star}$ and $\boldsymbol{\gamma}(d(t)) = \boldsymbol{s}_t$. We have

$$
\begin{aligned}
\left\langle P_{T_{\boldsymbol{s}_t} S}[\boldsymbol{s}^{\star}], \frac{-\log_{\boldsymbol{s}_t} \boldsymbol{s}^{\star}}{\|\log_{\boldsymbol{s}_t} \boldsymbol{s}^{\star}\|_2} \right\rangle &= \left\langle \boldsymbol{s}^{\star}, \frac{-\log_{\boldsymbol{s}_t}(\boldsymbol{s}^{\star})}{\|\log_{\boldsymbol{s}_t}(\boldsymbol{s}^{\star})\|_2} \right\rangle \\
&= \langle \boldsymbol{\gamma}(0), \dot{\boldsymbol{\gamma}}(d(t)) \rangle \\
&\leq -d(t) + \frac{1}{6} \kappa^2 d^3(t),
\end{aligned}
\tag{25}
$$

where we used the fact that $\dot{\boldsymbol{\gamma}}(d(t)) = \frac{-\log_{\boldsymbol{s}_t}(\boldsymbol{s}^{\star})}{\|\log_{\boldsymbol{s}_t}(\boldsymbol{s}^{\star})\|_2} \in T_{\boldsymbol{s}_t} S$, and the inequality is given by Lemma 3.

For the integrand of the second term in (23), we have

$$
\begin{aligned}
\left\langle P_{T_{\boldsymbol{s}_t} S}[\boldsymbol{x} - \boldsymbol{s}^{\star}], \frac{-\log_{\boldsymbol{s}_t} \boldsymbol{s}^{\star}}{\|\log_{\boldsymbol{s}_t} \boldsymbol{s}^{\star}\|_2} \right\rangle &\leq \|P_{T_{\boldsymbol{s}_t} S}[\boldsymbol{x} - \boldsymbol{s}^{\star}]\|_2 \\
&= \|P_{T_{\boldsymbol{s}_t} S} \, P_{(T_{\boldsymbol{s}^{\star}} S)^{\perp}}[\boldsymbol{x} - \boldsymbol{s}^{\star}]\|_2 \\
&\leq \|P_{T_{\boldsymbol{s}_t} S} \, P_{(T_{\boldsymbol{s}^{\star}} S)^{\perp}}\| \, \|\boldsymbol{x} - \boldsymbol{s}^{\star}\|_2 \\
&\leq \|P_{(T_{\boldsymbol{s}^{\star}} S)^{\perp}} \, P_{T_{\boldsymbol{s}_t} S}\| \, \|\boldsymbol{z}\|_2,
\end{aligned}
\tag{26}
$$

where we used the optimality of $\boldsymbol{s}^{\star}$ and the symmetry of projection operators. The operator norm $\|P_{(T_{\boldsymbol{s}^{\star}} S)^{\perp}} \, P_{T_{\boldsymbol{s}_t} S}\|$ can be rewritten as

$$
\|P_{(T_{\boldsymbol{s}^{\star}} S)^{\perp}} \, P_{T_{\boldsymbol{s}_t} S}\| = \sup_{\boldsymbol{v} \in T_{\boldsymbol{s}_t} S, \, \|\boldsymbol{v}\|_2 = 1} d(\boldsymbol{v}, T_{\boldsymbol{s}^{\star}} S).
\tag{27}
$$

For any unit vector $\boldsymbol{v} \in T_{\boldsymbol{s}_t} S$, we will construct a vector in $T_{\boldsymbol{s}^{\star}} S$ and use its distance from $\boldsymbol{v}$ to upper bound $d(\boldsymbol{v}, T_{\boldsymbol{s}^{\star}} S)$. We again use the unit-speed geodesic $\boldsymbol{\gamma}$ joining $\boldsymbol{s}^{\star}$ and $\boldsymbol{s}_t$, where

$\boldsymbol{\gamma}(0) = \boldsymbol{s}^\star$ and $\boldsymbol{\gamma}(d(t)) = \boldsymbol{s}_t$. Let $\boldsymbol{v}_r = \mathcal{P}_{r,d(t)}\boldsymbol{v}$ for $r \in [0, d(t)]$, where $\mathcal{P}_{r,d(t)}$ denotes the parallel transport backward along $\boldsymbol{\gamma}$. The derivative of $\boldsymbol{v}_r$ can be expressed by the second fundamental form $\frac{d}{dr}\boldsymbol{v}_r = \mathrm{I\!I}(\dot{\boldsymbol{\gamma}}(r), \boldsymbol{v}_r)$, which can further be bounded by Lemma 4 to get $\|\frac{d}{dr}\boldsymbol{v}_r\|_2 \leq 3\kappa$. Hence $\|\boldsymbol{v}_{d(t)} - \boldsymbol{v}_0\|_2 \leq 3\kappa d(t)$. Since $\boldsymbol{v}_{d(t)} = \boldsymbol{v}$ and $\boldsymbol{v}_0 \in T_{\boldsymbol{s}^\star}S$, it follows that $d(\boldsymbol{v}, T_{\boldsymbol{s}^\star}S) \leq 3\kappa d(t)$ for any unit vector $\boldsymbol{v} \in T_{\boldsymbol{s}_t}S$. Hence

$$\|P_{(T_{\boldsymbol{s}^\star}S)^\perp} P_{T_{\boldsymbol{s}_t}S}\| \leq 3\kappa \cdot d(t). \tag{28}$$

Since $\|\boldsymbol{z}\|_2 \leq 2\sigma\sqrt{D}$ with high probability, plugging these into (26), we have with high probability

$$\left\langle P_{T_{\boldsymbol{s}_t}S}[\boldsymbol{x} - \boldsymbol{s}^\star], \frac{-\log_{\boldsymbol{s}_t}\boldsymbol{s}^\star}{\|\log_{\boldsymbol{s}_t}\boldsymbol{s}^\star\|_2} \right\rangle \leq 6\sigma\kappa\sqrt{D} \cdot d(t). \tag{29}$$

The integrand of the third term in (23) can be bounded using the Riemannian Hessian. We have

$$\mathrm{grad}[f](\boldsymbol{s}_t) = \Pi_{\boldsymbol{s}_t, \boldsymbol{s}^k}\{\mathrm{grad}[f](\boldsymbol{s}^k)\} + \int_{r=0}^{t} \Pi_{\boldsymbol{s}_t, \boldsymbol{s}_r} \mathrm{Hess}[f](\boldsymbol{s}_r) \Pi_{\boldsymbol{s}_r, \boldsymbol{s}^k}\{-\mathrm{grad}[f](\boldsymbol{s}^k)\}\, dr. \tag{30}$$

Using the $L$-Lipschitz gradient property of the function $f$ from Lemma 5, we have

$$\left\langle \mathrm{grad}[f](\boldsymbol{s}_t) + \Pi_{\boldsymbol{s}_t, \boldsymbol{s}^k}\{-\mathrm{grad}[f](\boldsymbol{s}^k)\}, \frac{-\log_{\boldsymbol{s}_r}\boldsymbol{s}^\star}{\|\log_{\boldsymbol{s}_r}\boldsymbol{s}^\star\|_2} \right\rangle \leq \|\mathrm{grad}[f](\boldsymbol{s}_t) - \Pi_{\boldsymbol{s}_t, \boldsymbol{s}^k}\{\mathrm{grad}[f](\boldsymbol{s}^k)\}\|_2$$

$$\leq t \max_{\bar{s}} \|\mathrm{Hess}[f](\bar{s})\| \|\mathrm{grad}[f](\boldsymbol{s}^k)\|_2$$

$$\leq tL^2 d(\boldsymbol{s}^k, \boldsymbol{s}^\star). \tag{31}$$

Hence

$$\int_0^\tau \left\langle \mathrm{grad}[f](\boldsymbol{s}_t) + \Pi_{\boldsymbol{s}_t, \boldsymbol{s}^k}\{-\mathrm{grad}[f](\boldsymbol{s}^k)\}, \frac{-\log_{\boldsymbol{s}_r}\boldsymbol{s}^\star}{\|\log_{\boldsymbol{s}_r}\boldsymbol{s}^\star\|_2} \right\rangle dt \leq \frac{1}{2}\tau^2 L^2 d(\boldsymbol{s}^k, \boldsymbol{s}^\star). \tag{32}$$

Gathering the separate bounds of the three terms in (23), we have

$$d(\tau) \leq d(0) + \int_0^\tau \left( -d(t) + \frac{1}{6}\kappa^2 d^3(t) + 6\sigma\kappa\sqrt{D}d(t) \right) dt + \frac{1}{2}L^2\tau^2 d(0)$$

$$= (1 + \frac{1}{2}L^2\tau^2)d(0) + \int_0^\tau \left( -(1 - c_1)d(t) + \frac{1}{6}\kappa^2 d^3(t) \right) dt, \tag{33}$$

where $c_1 = 6\sigma\kappa\sqrt{D} \leq \frac{1}{10}$. By triangle inequality we have

$$d(\boldsymbol{s}^k, \boldsymbol{s}^\star) - d(\boldsymbol{s}^k, \boldsymbol{s}_t) \leq d(\boldsymbol{s}_t, \boldsymbol{s}^\star) \leq d(\boldsymbol{s}^k, \boldsymbol{s}^\star) + d(\boldsymbol{s}^k, \boldsymbol{s}_t). \tag{34}$$

Since $\boldsymbol{s}_t = \exp_{\boldsymbol{s}^k}\left( -t\,\mathrm{grad}[f](\boldsymbol{s}^k) \right)$, we have

$$d(\boldsymbol{s}^k, \boldsymbol{s}_t) \leq t\|\mathrm{grad}[f](\boldsymbol{s}^k)\|_2 \leq tL \cdot d(\boldsymbol{s}^k, \boldsymbol{s}^*), \tag{35}$$

and thus

$$(1 - tL)d(0) \leq d(t) \leq (1 + tL)d(0). \tag{36}$$

Hence for the integrand in (33), we have

$$-(1 - c_1)d(t) + \frac{1}{6}\kappa^2 d^3(t) \leq -(1 - c_1)d(t) + \frac{1}{6}\kappa^2(1 + tL)^2 d^2(0)d(t)$$

$$\leq -(1 - c_1)d(t) + \frac{1}{6}\kappa^2(1 + \tau L)^2(2\Delta)^2 d(t)$$

$$= (-1 + c_2)d(t)$$

$$\leq (-1 + c_2)(1 - tL)d(0) \tag{37}$$

where $c_2 = c_1 + \frac{2}{3}(1 + \tau L)^2$.

Plugging this back, we get

$$d(\tau) \leq (1 + \frac{1}{2}L^2\tau^2)d(0) + (-1 + c_2)d(0)\int_0^\tau (1 - tL)dt$$

$$= \left( 1 - (1 - c_2)\tau + \left( \frac{1}{2}L^2 + \frac{1}{2}L(1 - c_2) \right)\tau^2 \right)d(0). \tag{38}$$

Substituting in $L = \frac{121}{30}$ from Lemma 5 and $\tau = \frac{1}{64}$, we get

$$d(\boldsymbol{s}^{k+1}, \boldsymbol{s}^\star) \leq (1 - \epsilon)d(\boldsymbol{s}^k, \boldsymbol{s}^\star) \tag{39}$$

where $\epsilon \approx 2.3 \times 10^{-4}$, which proves result (10). Note that this also implies the uniqueness of the minimizer $\boldsymbol{s}^\star$.

$\square$

### C.1 SUPPORTING LEMMAS

**Lemma 3.** *Let $\boldsymbol{\gamma}$ be a regular unit-speed curve on the manifold $S \subset \mathbb{S}^{d-1}$ with extrinsic curvature $\kappa$. Then,*

$$\langle \dot{\boldsymbol{\gamma}}(t), \boldsymbol{\gamma}(0) \rangle \leq -t + \frac{\kappa^2 t^3}{6} \tag{40}$$

*Proof.* Since $\boldsymbol{\gamma} \subset \mathbb{S}^{d-1}$, by differentiating both sides of $\|\boldsymbol{\gamma}(t)\|_2^2 = 1$ we get $\langle \dot{\boldsymbol{\gamma}}(t), \boldsymbol{\gamma}(t) \rangle = 0$. Further, since $\boldsymbol{\gamma}$ is unit-speed, we have $\|\dot{\boldsymbol{\gamma}}(t)\|_2^2 = 1$ and by differentiating it $\langle \ddot{\boldsymbol{\gamma}}(t), \dot{\boldsymbol{\gamma}}(t) \rangle = 0$. Therefore,

$$
\begin{aligned}
\langle \dot{\boldsymbol{\gamma}}(t), \boldsymbol{\gamma}(0) \rangle &= \left\langle \dot{\boldsymbol{\gamma}}(t), \boldsymbol{\gamma}(t) - \int_0^t \dot{\boldsymbol{\gamma}}(t_1)\, dt_1 \right\rangle \\
&= -\left\langle \dot{\boldsymbol{\gamma}}(t), \int_0^t \dot{\boldsymbol{\gamma}}(t_1)\, dt_1 \right\rangle \\
&= -\int_{t_1=0}^t \left\langle \dot{\boldsymbol{\gamma}}(t), \dot{\boldsymbol{\gamma}}(t) - \int_{t_2=t_1}^t \ddot{\boldsymbol{\gamma}}(t_2)\, dt_2 \right\rangle dt_1 \\
&= -t + \int_{t_1=0}^t \int_{t_2=t_1}^t \langle \dot{\boldsymbol{\gamma}}(t), \ddot{\boldsymbol{\gamma}}(t_2) \rangle\, dt_2\, dt_1 \\
&= -t + \int_{t_1=0}^t \int_{t_2=t_1}^t \left\langle \dot{\boldsymbol{\gamma}}(t_2) + \int_{t_3=t_2}^t \ddot{\boldsymbol{\gamma}}(t_3)\, dt_3, \ddot{\boldsymbol{\gamma}}(t_2) \right\rangle dt_2\, dt_1 \\
&= -t + \int_{t_1=0}^t \int_{t_2=t_1}^t \int_{t_3=t_2}^t \langle \ddot{\boldsymbol{\gamma}}(t_3), \ddot{\boldsymbol{\gamma}}(t_2) \rangle\, dt_3\, dt_2\, dt_1 \\
&\leq -t + \kappa^2 \int_{t_1=0}^t \int_{t_2=t_1}^t \int_{t_3=t_2}^t dt_3\, dt_2\, dt_1 \\
&= -t + \frac{\kappa^2 t^3}{6}.
\end{aligned}
\tag{41}
$$

$\square$

**Lemma 4.** *Let $\mathbb{II}(\boldsymbol{u}, \boldsymbol{v})$ denote the second fundamental form at some point $\boldsymbol{s} \in S$, and let $\kappa$ denote the extrinsic ($\mathbb{R}^D$) geodesic curvature of $S$. Then*

$$\sup_{\|\boldsymbol{u}\|_2=1, \|\boldsymbol{v}\|_2=1} \|\mathbb{II}(\boldsymbol{u}, \boldsymbol{v})\|_2 \leq 3\kappa. \tag{42}$$

*Proof.* Set

$$\kappa^{\mathbb{II}} = \max_{\|\boldsymbol{u}\|_2=1, \|\boldsymbol{v}\|_2=1} \|\mathbb{II}(\boldsymbol{u}, \boldsymbol{v})\|_2^2. \tag{43}$$

Choose unit vectors $\boldsymbol{u}$, $\boldsymbol{v}$ which realize this maximum value (these must exist, by continuity of $\mathbb{II}$ and compactness of the constraint set). Because $\mathbb{II}$ is bilinear, $\|\mathbb{II}(\boldsymbol{u}, \boldsymbol{v})\|_2^2 = \|\mathbb{II}(\boldsymbol{u}, -\boldsymbol{v})\|_2^2$, and without loss of generality, we can assume $\langle \boldsymbol{u}, \boldsymbol{v} \rangle \leq 0$.

Since $\mathbb{II}(\boldsymbol{u}, \boldsymbol{v})$ is a symmetric bilinear form, each coordinate of the vector $\mathbb{II}(\boldsymbol{u}, \boldsymbol{v})$ has the form $\mathbb{II}_i(\boldsymbol{u}, \boldsymbol{v}) = \boldsymbol{u}^T \boldsymbol{\Phi}_i \boldsymbol{v}$ for some symmetric $d \times d$ matrix $\boldsymbol{\Phi}_i$. Now,

$$\boldsymbol{u}^T \boldsymbol{\Phi}_i \boldsymbol{v} = \tfrac{1}{2}(\boldsymbol{u} + \boldsymbol{v})^T \boldsymbol{\Phi}_i (\boldsymbol{u} + \boldsymbol{v}) - \tfrac{1}{2}\boldsymbol{u}^T \boldsymbol{\Phi}_i \boldsymbol{u} - \tfrac{1}{2}\boldsymbol{v}^T \boldsymbol{\Phi}_i \boldsymbol{v}, \tag{44}$$

so

$$|\tfrac{1}{2}\boldsymbol{u}^T\boldsymbol{\Phi}_i\boldsymbol{u}| + |\tfrac{1}{2}\boldsymbol{v}^T\boldsymbol{\Phi}_i\boldsymbol{v}| + |\tfrac{1}{2}(\boldsymbol{u}+\boldsymbol{v})^T\boldsymbol{\Phi}_i(\boldsymbol{u}+\boldsymbol{v})| \geq |\boldsymbol{u}^T\boldsymbol{\Phi}_i\boldsymbol{v}| \tag{45}$$

and

$$3|\tfrac{1}{2}\boldsymbol{u}^T\boldsymbol{\Phi}_i\boldsymbol{u}|^2 + 3|\tfrac{1}{2}\boldsymbol{v}^T\boldsymbol{\Phi}_i\boldsymbol{v}|^2 + 3|\tfrac{1}{2}(\boldsymbol{u}+\boldsymbol{v})^T\boldsymbol{\Phi}_i(\boldsymbol{u}+\boldsymbol{v})|^2 \geq |\boldsymbol{u}^T\boldsymbol{\Phi}_i\boldsymbol{v}|^2 \tag{46}$$

where we have used the inequality $(a + b + c)^2 \leq 3a^2 + 3b^2 + 3c^2$ which follows from convexity of the square. Summing over $i$, we obtain that

$$\tfrac{3}{4}\|\mathbb{I}(\boldsymbol{u},\boldsymbol{u})\|_2^2 + \tfrac{3}{4}\|\mathbb{I}(\boldsymbol{v},\boldsymbol{v})\|_2^2 + \tfrac{3}{4}\|\mathbb{I}(\boldsymbol{u}+\boldsymbol{v},\boldsymbol{u}+\boldsymbol{v})\|_2^2 \geq \|\mathbb{I}(\boldsymbol{u},\boldsymbol{v})\|_2^2 \tag{47}$$

this implies that

$$\tfrac{9}{4}\max\Big\{\|\mathbb{I}(\boldsymbol{u},\boldsymbol{u})\|_2^2, \|\mathbb{I}(\boldsymbol{v},\boldsymbol{v})\|_2^2, \|\mathbb{I}(\boldsymbol{u}+\boldsymbol{v},\boldsymbol{u}+\boldsymbol{v})\|_2^2\Big\} \geq \|\mathbb{I}(\boldsymbol{u},\boldsymbol{v})\|_2^2 \tag{48}$$

Because $\boldsymbol{u}, \boldsymbol{v}$ are unit vectors with $\langle \boldsymbol{u}, \boldsymbol{v}\rangle \leq 0$, we have $\|\boldsymbol{u}+\boldsymbol{v}\|_2 \leq \sqrt{2}$, and so

$$4\kappa^2 \geq \max\Big\{\|\mathbb{I}(\boldsymbol{u},\boldsymbol{u})\|_2^2, \|\mathbb{I}(\boldsymbol{v},\boldsymbol{v})\|_2^2, \|\mathbb{I}(\boldsymbol{u}+\boldsymbol{v},\boldsymbol{u}+\boldsymbol{v})\|_2^2\Big\}, \tag{49}$$

whence

$$9\kappa^2 \geq \|\mathbb{I}(\boldsymbol{u},\boldsymbol{v})\|_2^2 = (\kappa^{\mathbb{I}})^2, \tag{50}$$

which is the claimed inequality. $\qquad\square$

**Lemma 5.** *Assume $\sigma \leq 1/(60\kappa\sqrt{D})$. The objective function $f(\boldsymbol{s}) = -\langle \boldsymbol{s}, \boldsymbol{x}\rangle$ has $L-$Lipschitz gradient in a $1/\kappa$-neighborhood of $\boldsymbol{s}_\natural$ with probability at least $1 - e^{-D/2}$, where $L = \frac{121}{30}$.*

*Proof.* On a Riemannian manifold $S$, the conditions for $L$-Lipschitz gradient in a subset can be expressed as $\frac{d^2}{dt^2}(f \circ \boldsymbol{\gamma})(t) \leq L$ for all unit-speed geodesics $\boldsymbol{\gamma}(t)$ in the subset Boumal (2023).

Let $\Delta = 1/\kappa$, and let $\boldsymbol{\gamma}(t)$ be a unit-speed geodesic of $S$ in the neighborhood $B(\boldsymbol{s}_\natural, \Delta)$, $t \in [0, T]$. The neighborhood constraint implies that $T = d(\boldsymbol{\gamma}(0), \boldsymbol{\gamma}(T)) \leq d(\boldsymbol{\gamma}(0), \boldsymbol{s}_\natural) + d(\boldsymbol{\gamma}(T), \boldsymbol{s}_\natural) \leq 2\Delta$.

To bound the second derivative $\frac{d^2}{dt^2}(f \circ \boldsymbol{\gamma})(t)$, we have

$$\begin{aligned}
\frac{d^2}{dt^2}(f \circ \boldsymbol{\gamma})(t) &= -\langle \ddot{\boldsymbol{\gamma}}(t), \boldsymbol{x}\rangle \\
&= -\langle \ddot{\boldsymbol{\gamma}}(t), \boldsymbol{\gamma}(0)\rangle - \langle \ddot{\boldsymbol{\gamma}}(t), \boldsymbol{s}_\natural - \boldsymbol{\gamma}(0)\rangle - \langle \ddot{\boldsymbol{\gamma}}(t), \boldsymbol{z}\rangle.
\end{aligned} \tag{51}$$

The first term can be bounded as

$$\begin{aligned}
-\langle \ddot{\boldsymbol{\gamma}}(t), \boldsymbol{\gamma}(0)\rangle &= -\Big\langle \ddot{\boldsymbol{\gamma}}(t), \boldsymbol{\gamma}(t) - \int_{t_1=0}^{t} \dot{\boldsymbol{\gamma}}(t_1)dt_1 \Big\rangle \\
&= -\langle \ddot{\boldsymbol{\gamma}}(t), \boldsymbol{\gamma}(t)\rangle + \int_{t_1=0}^{t} \langle \ddot{\boldsymbol{\gamma}}(t), \dot{\boldsymbol{\gamma}}(t_1)\rangle \, dt_1 \\
&= 1 + \int_{t_1=0}^{t} \Big\langle \ddot{\boldsymbol{\gamma}}(t), \dot{\boldsymbol{\gamma}}(t) - \int_{t_2=t_1}^{t} \ddot{\boldsymbol{\gamma}}(t_2)dt_2 \Big\rangle dt_1 \\
&= 1 - \int_{t_1=0}^{t}\int_{t_2=t_1}^{t} \langle \ddot{\boldsymbol{\gamma}}(t), \ddot{\boldsymbol{\gamma}}(t_2)\rangle \, dt_2 dt_1 \\
&\leq 1 + \kappa^2 \int_{t_1=0}^{t}\int_{t_2=t_1}^{t} dt_2 dt_1 \\
&\leq 1 + \frac{1}{2}\kappa^2 T^2 \\
&\leq 1 + 2\kappa^2\Delta^2,
\end{aligned} \tag{52}$$

where we used $\langle \ddot{\boldsymbol{\gamma}}(t), \boldsymbol{\gamma}(t)\rangle = -1$ (by differentiating both sides of $\langle \dot{\boldsymbol{\gamma}}(t), \boldsymbol{\gamma}(t)\rangle = 0$) and $\langle \ddot{\boldsymbol{\gamma}}(t), \dot{\boldsymbol{\gamma}}(t)\rangle = 0$.

Hence

$$\frac{d^2}{dt^2}(f \circ \boldsymbol{\gamma})(t) \leq 1 + 2\kappa^2\Delta^2 + \kappa\Delta + \kappa\|\boldsymbol{z}\|. \tag{53}$$

Since $\|\boldsymbol{z}\|_2 \leq 2\sigma\sqrt{D}$ with probability at least $1 - e^{-D/2}$, combining this with $\Delta = 1/\kappa$ and $\sigma \leq \frac{1}{60\kappa\sqrt{D}}$, we get with high probability $\frac{d^2}{dt^2}(f \circ \boldsymbol{\gamma})(t) \leq \frac{121}{30}$. $\qquad\square$

# D   CHAINING BOUNDS FOR THE TANGENT BUNDLE PROCESS

In this section, we prove the following lemma, which bounds a crucial Gaussian process that arises in the analysis of gradient descent.

---

**Main Bound for Tangent Bundle Process**

**Theorem 6.** *Suppose that $\Delta \leq 1/\kappa$, and set*

$$T^{\mathrm{max}} = \sup\Big\{ \langle \boldsymbol{v}, \boldsymbol{z} \rangle \mid d_S(\boldsymbol{s}, \boldsymbol{s}_\natural) \leq \Delta, \ \boldsymbol{v} \in T_{\boldsymbol{s}}S, \ \|\boldsymbol{v}\|_2 = 1 \Big\}. \tag{54}$$

*Then with probability at least $1 - 1.6e^{-\frac{x^2}{2\sigma^2}}$, we have*

$$T^{\mathrm{max}} \leq 12\sigma(\kappa\sqrt{2\pi(d+1)} + \sqrt{\log 12\kappa}) + 30x. \tag{55}$$

---

We prove Theorem 6 below. We directly follow the proof of Theorem 5.29 from Van Handel (2016), establishing a chaining argument while accounting for slight discrepancies and establishing exact constants. The main geometric content of this argument is in Lemma 9, which bounds the size of $\varepsilon$-nets for the tangent bundle.

*Proof.* Set

$$\mathcal{V} = \Big\{ \boldsymbol{v} \mid \boldsymbol{v} \in T_{\boldsymbol{s}}S, \ \|\boldsymbol{v}\|_2 = 1, \ d_S(\boldsymbol{s}, \boldsymbol{s}_\natural) \leq \Delta \Big\}, \tag{56}$$

We first prove that $\mathcal{T} = \{\langle \boldsymbol{v}, \boldsymbol{z} \rangle\}_{v \in \mathcal{V}}$ defines a separable, sub-gaussian process. Take any $v, v' \in \mathcal{V}$. Then

$$\langle \boldsymbol{v}, \boldsymbol{z} \rangle - \langle \boldsymbol{v}', \boldsymbol{z} \rangle = \langle \boldsymbol{v} - \boldsymbol{v}', \boldsymbol{z} \rangle \sim \mathcal{N}(0, \sigma^2 d(\boldsymbol{v}, \boldsymbol{v}')^2), \tag{57}$$

immediately satisfying sub-gaussianity. By Lemma 9, there exists an $\varepsilon$-net $\mathcal{N}(\mathcal{V}, d, \epsilon)$ for $\mathcal{V}$ of size at most $N = (12\kappa/\varepsilon)^{2d+1}$. To see separability, let $\mathcal{N}_k = \mathcal{N}(\mathcal{V}, d, 2^{-k})$ be the epsilon net corresponding to $\epsilon = \dfrac{1}{2^k}$. We can construct a countable dense subset of $\mathcal{V}$ by letting

$$\mathcal{N}_\infty = \bigcup_{k=1}^{\infty} \mathcal{N}(\mathcal{V}, d, 2^{-k}). \tag{58}$$

Therefore, the existence of a countable dense subset implies separability of $\mathcal{V}$ immediately implying separability of $\mathcal{T}$. Using these facts, we first prove the result in the finite case $|\mathcal{V}| < \infty$, after which we use separability to extend to the infinite case.

Let $|\mathcal{V}| < \infty$ and $k_0$ be the largest integer such that $2^{-k_0} \geq \mathrm{diam}(\mathcal{V})$. Define $\mathcal{N}_{k_0} = \mathcal{N}(\mathcal{V}, d, 2^{-k_0})$ to be a $2^{-k_0}$ net of $\mathcal{V}$ with respect to the metric $d$. Then for all $\boldsymbol{v} \in \mathcal{V}$, there exists $\pi_0(\boldsymbol{v}) \in \mathcal{N}_{k_0}$ such that $d(\boldsymbol{v}, \pi_0(\boldsymbol{v})) < 2^{-k_0}$.

For $k > k_0$, let $\mathcal{N}_k = \mathcal{N}(\mathcal{V}, d, 2^{-k})$ be a $2^{-k}$ net of $\mathcal{V}$. Subsequently for all $\boldsymbol{v} \in \mathcal{V}$, there exists $\pi_k(\boldsymbol{v}) \in \mathcal{N}_{k_0}$ such that $d(\boldsymbol{v}, \pi_k(\boldsymbol{v})) < 2^{-k}$.

Now fix any $\boldsymbol{v_0} \in \mathcal{V}$. For any $\boldsymbol{v} \in \mathcal{V}$, sufficiently large $n$ yields $\pi_n(\boldsymbol{v}) = \boldsymbol{v}$. Thus,

$$\langle \boldsymbol{v}, \boldsymbol{z} \rangle - \langle \boldsymbol{v_0}, \boldsymbol{z} \rangle = \sum_{k > k_0} \{\langle \pi_k(\boldsymbol{v}), \boldsymbol{z} \rangle - \langle \pi_{k-1}(\boldsymbol{v}), \boldsymbol{z} \rangle\} \tag{59}$$

by the telescoping property, implying

$$\sup_{\boldsymbol{v} \in \mathcal{T}}\{\langle \boldsymbol{v}, \boldsymbol{z} \rangle - \langle \boldsymbol{v_0}, \boldsymbol{z} \rangle\} \leq \sum_{k > k_0} \sup_{\boldsymbol{v} \in \mathcal{V}}\{\langle \pi_k(\boldsymbol{v}), \boldsymbol{z} \rangle - \langle \pi_{k-1}(\boldsymbol{v}), \boldsymbol{z} \rangle\}. \tag{60}$$

Using the fact that $\mathcal{T}$ is a sub-gaussian process and Lemma 5.2 of Van Handel (2016), we can bound each individual sum as

$$\mathbb{P}(\sup_{\boldsymbol{v}\in\mathcal{V}}\{\langle\pi_k(\boldsymbol{v}),\boldsymbol{z}\rangle - \langle\pi_{k-1}(\boldsymbol{v}),\boldsymbol{z}\rangle\} \geq 6\times 2^{-k}\sigma\sqrt{\log|\mathcal{N}_k|} + 3\times 2^{-k}x_k) \leq e^{-\frac{x_k^2}{2\sigma^2}}. \tag{61}$$

By ensuring that all of the sums are simultaneously controlled, we can arrive at the desired bound. We first derive the complement (i.e. there exists one sum which exceeds the desired value)

$$\mathbb{P}(A^c) := \mathbb{P}(\exists k > k_0 \text{ s.t. } \sup_{\boldsymbol{v}\in\mathcal{V}}\{\langle\pi_k(\boldsymbol{v}),\boldsymbol{z}\rangle - \langle\pi_{k-1}(\boldsymbol{v}),\boldsymbol{z}\rangle\} \geq 6\times 2^{-k}\sigma\sqrt{\log|N_k|} + 3\times 2^{-k}x_k) \tag{62}$$

$$\leq \sum_{k>k_0}\mathbb{P}(\sup_{\boldsymbol{v}\in\mathcal{V}}\{\langle\pi_k(\boldsymbol{v}),\boldsymbol{z}\rangle - \langle\pi_{k-1}(\boldsymbol{v}),\boldsymbol{z}\rangle\} \geq 6\times 2^{-k}\sigma\sqrt{\log|N_k|} + 3\times 2^{-k}x_k) \tag{63}$$

$$\leq \sum_{k>k_0}e^{-\frac{x_k^2}{2\sigma^2}} \tag{64}$$

$$\leq e^{-\frac{x^2}{2\sigma^2}}\sum_{k>0}e^{-k/2} \leq 1.6e^{-\frac{x^2}{2\sigma^2}} \tag{65}$$

Now, using corollary 5.25 of Van Handel (2016) and $|\mathcal{N}| \leq (\frac{12\kappa}{\epsilon})^{2d+1}$, we have

$$\sup_{\boldsymbol{v}\in V}\{\langle\boldsymbol{v},\boldsymbol{z}\rangle - \langle\boldsymbol{v_0},\boldsymbol{z}\rangle\} \leq \sum_{k>k_0}\sup_{\boldsymbol{v}\in\mathcal{V}}\{\langle\pi_k(\boldsymbol{v}),\boldsymbol{z}\rangle - \langle\pi_{k-1}(\boldsymbol{v}),\boldsymbol{z}\rangle\} \tag{66}$$

$$\leq 6\sum_{k>k_0}2^{-k}\sigma\sqrt{\log|\mathcal{N}_k|} + 3\times 2^{-k_0}\sum_{k>0}2^{-k}\sqrt{k} + 3\times 2^{-k_0}\sum_{k>0}2^{-k}x \tag{67}$$

$$\leq 12\int_0^\infty \sigma\sqrt{\log\mathcal{N}(\mathcal{V},d,\epsilon)}\,d\epsilon + 15\mathrm{diam}(\mathcal{V})x \tag{68}$$

$$\leq 12\sigma\sqrt{2d+1}\int_0^\infty \sqrt{(\log(\frac{12\kappa}{\epsilon})}\,d\epsilon + 15\mathrm{diam}(\mathcal{V})x \tag{69}$$

$$= 12\sigma\sqrt{2d+1}(\kappa\sqrt{\pi}\,\mathrm{erf}\,(\log 12\kappa) + \sqrt{\log 12\kappa}) + 15\mathrm{diam}(\mathcal{V})x \tag{70}$$

$$\leq 12\sigma(\kappa\sqrt{2\pi(d+1)} + \sqrt{\log 12\kappa}) + 15\mathrm{diam}(\mathcal{V})x, \tag{71}$$

where we have used $2^{-k_0} \leq 2\mathrm{diam}(\mathcal{V})$, $\sum_{k>0}2^{-k}\sqrt{k} \leq 1.35$ and $\sum_{k>0}2^{-k} \leq 1$ in (67), and erf $z = \frac{2}{\sqrt{\pi}}\int_0^z e^{-t^2/2}dt \leq 1$ in (71).

Thus, if $A$ occurs the above equation holds, implying

$$\mathbb{P}[\sup_{\boldsymbol{v}\in\mathcal{V}}\{\langle\boldsymbol{v},\boldsymbol{z}\rangle - \langle\boldsymbol{v_0},\boldsymbol{z}\rangle\} \geq 12\sigma(\kappa\sqrt{2\pi(d+1)} + \sqrt{\log 12\kappa}) + 15\mathrm{diam}(\mathcal{V})x] \leq \mathbb{P}(A^c) \leq 1.6e^{-\frac{x^2}{2\sigma^2}} \tag{72}$$

Since $\mathcal{T}$ is a separable process, Theorem 5.24 of Van Handel (2016) directly extends the result to infinite/uncountable $\mathcal{T}$. Letting $\langle\boldsymbol{v_0},\boldsymbol{z}\rangle = 0$ and noting $\mathrm{diam}(\mathcal{V}) = \sup_{\boldsymbol{v},\boldsymbol{v}'\in\mathcal{V}}||\boldsymbol{v}-\boldsymbol{v}'||_2 \leq 2$ yields the claim. $\qquad\square$

NETS FOR $B(\boldsymbol{s}_\natural,\Delta)$

**Lemma 7.** *Suppose that $\Delta < 1/\kappa$. For any $\varepsilon \in (0,...]$, there exists an $\varepsilon$-net $\widehat{S}$ for $B(\boldsymbol{s}_\natural,\Delta)$ of size $\#\widehat{S} < (12/\varepsilon)^{d+1}$.*

At a high level, the proof of this lemma proceeds as follows: we form an $\varepsilon_0$ net $N_0$ for $T_{\boldsymbol{s}_\natural}S$, and then set $\widehat{S} = \{\exp_{\boldsymbol{s}_\natural}(\boldsymbol{v}) \mid \boldsymbol{v}\in N_0\}$. We will argue that $\widehat{S}$ is a $C\varepsilon_0$-net for $B(\boldsymbol{s}_\natural,\Delta)$, by arguing that

at length scales $\Delta < 1/\kappa$, the distortion induced by the exponential map is bounded. Crucial to this argument is the following lemma on geodesic triangles:

**Lemma 8.** *Consider $\boldsymbol{v}, \boldsymbol{v}' \in T_{\boldsymbol{s}_\natural}S$, with $\|\boldsymbol{v}\|_2 = \|\boldsymbol{v}'\|_2 < \Delta$. Then if $\angle(\boldsymbol{v}, \boldsymbol{v}') < \frac{1}{\sqrt{3}}$,*

$$d_S\left(\exp_{\boldsymbol{s}_\natural}(\boldsymbol{v}), \exp_{\boldsymbol{s}_\natural}(\boldsymbol{v}')\right) \leq \sqrt{6}\,\Delta\,\angle(\boldsymbol{v}, \boldsymbol{v}'). \tag{73}$$

This lemma says that the third side of the triangle with vertices $\boldsymbol{s}_\natural, \exp_{\boldsymbol{s}_\natural}(\boldsymbol{v}), \exp_{\boldsymbol{s}_\natural}(\boldsymbol{v}')$ is at most a constant longer than the third side of an analogous triangle in Euclidean space. The proof of this is a direct application of Toponogov's theorem, a fundamental result in Riemannian geometry which allows one to compare triangles in an arbitrary Riemannian manifold whose sectional curvature is lower bounded to triangles in a constant curvature model space, where one can apply concrete trigonometric reasoning.

**Proof of Lemma 8.** By Lemma 10, the sectional curvatures $\kappa_s$ of $S$ are uniformly bounded in terms of the extrinsic geodesic curvature $\kappa$:

$$\kappa_s \geq -\kappa^2. \tag{74}$$

By Toponogov's theorem Toponogov (2006), the length $d_S\left(\exp_{\boldsymbol{s}_\natural}(\boldsymbol{v}), \exp_{\boldsymbol{s}_\natural}(\boldsymbol{v}')\right)$, of the third side of the geodesic triangle $\boldsymbol{s}_\natural, \exp_{\boldsymbol{s}_\natural}(\boldsymbol{v}), \exp_{\boldsymbol{s}_\natural}(\boldsymbol{v}')$ is bounded by the length of the third side of a geodesic triangles with two sides of length $r = \|\boldsymbol{v}\| = \|\boldsymbol{v}'\|$ and angle $\theta = \angle(\boldsymbol{v}, \boldsymbol{v}')$ in the constant curvature model space $M_{-\kappa^2}$. We can rescale, so that this third length is bounded by $L/\kappa$, where $L$ is the length of the third side of a geodesic triangle with two sides of length $r\kappa$ and an angle of $\angle(\boldsymbol{v}, \boldsymbol{v}')$, in the hyperbolic space $M_{-1}$. Using hyperbolic trigonometry (cf Fact 11 and the identity $\cosh^2 t - \sinh^2 t = 1$), we have

$$\cosh L = 1 + \sinh^2(\kappa r) \times \left(1 - \cos\theta\right). \tag{75}$$

By convexity of sinh over $[0, \infty)$, for $t \in [0, 1]$, we have $\sinh(t) \leq t\sinh(1)$, and $\sinh^2(t) \leq t^2\sinh^2(1) < \frac{3}{2}t^2$; since $\kappa r < 1$, $\sinh^2(\kappa r) < \frac{3}{2}\kappa^2 r^2$. Since $\cos(t) \geq 1 - t^2$ for all $t$, we have

$$\cosh L \leq 1 + \tfrac{3}{2}\kappa^2 r^2 \theta^2. \tag{76}$$

Using $\kappa r < 1$, for $\theta < \frac{1}{\sqrt{3}}$ we have $\cosh(L) \leq \frac{3}{2} < \cosh(1)$. Noting that for $t \in [0, 1]$,

$$\cosh(t) \geq g(t) = 1 + \tfrac{1}{4}t^2, \tag{77}$$

on $s \in [0, \cosh(1)]$, we have $\cosh^{-1}(s) \leq g^{-1}(s) = 2\sqrt{s - 1}$, giving

$$L \leq \sqrt{6} \cdot \kappa r \theta. \tag{78}$$

Dividing by $\kappa$ gives the claimed bound. $\qquad\square$

**Proof of Lemma 7.** Form an (angular) $\varepsilon_0$-net $N_0$ for $\{\boldsymbol{v} \in T_{\boldsymbol{s}_\natural}S \mid \|\boldsymbol{v}\|_2 = 1\}$ satisfying

$$\forall\ \boldsymbol{v} \in T_{\boldsymbol{s}_\natural}S,\ \exists\,\widehat{\boldsymbol{v}} \in N_0 \text{ with } \angle(\boldsymbol{v}, \widehat{\boldsymbol{v}}) \leq \varepsilon, \tag{79}$$

and an $\varepsilon_0$-net

$$N_r = \{0, \varepsilon_0, 2\varepsilon_0, \ldots, \lfloor \Delta/\varepsilon_0 \rfloor\} \tag{80}$$

for the interval $[0, \Delta]$. We can take $\#N_0 \leq (3/\varepsilon_0)^d$ and $\#N_r \leq \Delta/\varepsilon_0 \leq 1/\varepsilon_0$. Combine these two to form a net $N$ for $\{\boldsymbol{v} \in T_{\boldsymbol{s}_0}S \mid \|\boldsymbol{v}\|_2 \leq \Delta\}$ by setting

$$N = \bigcup_{r \in N_r} rN_0. \tag{81}$$

Note that $\#N \leq (3/\varepsilon_0)^{d+1}$. Let $\widehat{S} = \{\exp_{\boldsymbol{s}_\natural}(\boldsymbol{v}) \mid \boldsymbol{v} \in N\}$. Consider an arbitrary element $\boldsymbol{s}$ of $B(\boldsymbol{s}_\natural, \Delta)$. There exists $\boldsymbol{v} \in T_{\boldsymbol{s}_\natural}S$ such that $\exp_{\boldsymbol{s}_\natural}(\boldsymbol{v}) = \boldsymbol{s}$. Set

$$\bar{\boldsymbol{v}} = \varepsilon_0 \left\lfloor \frac{\|\boldsymbol{v}\|_2}{\varepsilon_0} \right\rfloor \boldsymbol{v}. \tag{82}$$

There exists $\widehat{v} \in N$ with $\|\widehat{v}\|_2 = \|\bar{v}\|_2$ and $\angle(\widehat{v}, \bar{v}) \le \varepsilon_0$. Note that

$$\widehat{s} = \exp_{s_\natural}(\widehat{v}) \in \widehat{S}. \tag{83}$$

By Lemma 8, we have

$$
\begin{aligned}
d_S\left(s, \widehat{s}\right) &\le d_S\left(s, \exp_{s_\natural}(\bar{v})\right) + d_S\left(\exp_{s_\natural}(\bar{v}), \widehat{s}\right) \\
&\le \varepsilon_0 + 3\Delta\varepsilon_0 \\
&< 4\varepsilon_0.
\end{aligned}
\tag{84}
$$

Setting $\varepsilon_0 = \varepsilon/4$, we obtain that $\widehat{S}$ is an $\varepsilon$-net for $B(s_\natural, \Delta)$. $\qquad\square$

NETS FOR THE TANGENT BUNDLE

**Lemma 9.** *Set*

$$T = \left\{ v \mid v \in T_s S, \ \|v\|_2 = 1, \ d_S(s, s_\natural) \le \Delta \right\}, \tag{85}$$

*Then there exists an $\varepsilon$-net $\widehat{T}$ for $T$ of size*

$$\#\widehat{T} \le \left(\frac{12\kappa}{\varepsilon}\right)^{2d+1}. \tag{86}$$

*Proof.* Let $\widehat{S}$ be the $\varepsilon_0$-net for $B(s_\natural, \Delta)$. By Lemma 7, there exists such a net of size at most $(12/\varepsilon_0)^{d+1}$. For each $\widehat{s} \in \widehat{S}$, form an $\varepsilon_1$-net $N_{\widehat{s}}$ for

$$\left\{ v \in T_{\widehat{s}} S \mid \|v\|_2 = 1 \right\}. \tag{87}$$

We set

$$\widehat{T} = \bigcup_{\widehat{s} \in \widehat{S}} N_{\widehat{s}}. \tag{88}$$

By ? Lemma 5.2, we can take $\#N_{\widehat{s}} \le (3/\varepsilon_1)^d$, and so

$$\#\widehat{T} \le \left(\frac{3}{\varepsilon_1}\right)^d \left(\frac{12}{\varepsilon_0}\right)^{d+1}. \tag{89}$$

Consider an arbitrary element $v \in T$. The vector $v$ belongs to the tangent space $T_s S$ for some $s$. By construction, there exists $\widehat{s} \in \widehat{S}$ with $d_S(s, \widehat{s}) \le \varepsilon$. Consider a minimal geodesic $\gamma$ joining $s$ and $\widehat{s}$. We generate $\bar{v} \in T_{\widehat{s}} S$ by parallel transporting $v$ along $\gamma$. Let $\mathcal{P}_{t,0}$ denote this parallel transport. By ? Lemma 8.5, the vector field $v_t = \mathcal{P}_{t,0} v$ satisfies

$$\frac{d}{dt} v_t = \mathrm{I\!I}\left(\dot{\gamma}(s), v_t\right), \tag{90}$$

where $\mathrm{I\!I}(\cdot, \cdot)$ is the second fundamental form. So,

$$\mathcal{P}_{t,0} v = v + \int_0^t \mathrm{I\!I}\left(\dot{\gamma}(s), v_s\right) ds. \tag{91}$$

By Lemma 4, for every $s$

$$\left\|\mathrm{I\!I}\left(\dot{\gamma}(s), v_s\right)\right\| \le 3\kappa \tag{92}$$

and

$$\|\bar{v} - v\| \le 3\varepsilon_0 \kappa. \tag{93}$$

By construction, there is an element $\widehat{v}$ of $N_{\widehat{s}}$ with

$$\|\widehat{v} - \bar{v}\| \le \varepsilon_1, \tag{94}$$

and so $\widehat{T}$ is an $\varepsilon_1 + 3\kappa\varepsilon_0$-net for $T$. Setting $\varepsilon_1 = \varepsilon/4$ and $\varepsilon_0 = \varepsilon/4\kappa$ completes the proof. $\qquad\square$

SUPPORTING RESULTS ON GEOMETRY

**Lemma 10.** *For a Riemannian submanifold $S$ of $\mathbb{R}^D$, the sectional curvatures $\kappa_s(\boldsymbol{v}, \boldsymbol{v}')$ are bounded by the extrinsic geodesic curvature $\kappa$, as*

$$\kappa_s(\boldsymbol{v}, \boldsymbol{v}') \geq -\kappa^2. \tag{95}$$

*Proof.* Using the Gauss formula (Theorem 8.4 of **?**), the Riemann curvature tensor $R_S$ of $S$ is related to the Riemann curvature tensor $R_{\mathbb{R}^n}$ of the ambient space via

$$
\begin{aligned}
\langle R_S(\boldsymbol{u}, \boldsymbol{v})\boldsymbol{v}, \boldsymbol{u}\rangle &= \langle R_{\mathbb{R}^D}(\boldsymbol{u}, \boldsymbol{v})\boldsymbol{v}, \boldsymbol{u}\rangle + \langle \mathbb{II}(\boldsymbol{u}, \boldsymbol{v}), \mathbb{II}(\boldsymbol{u}, \boldsymbol{v})\rangle - \langle \mathbb{II}(\boldsymbol{u}, \boldsymbol{u}), \mathbb{II}(\boldsymbol{v}, \boldsymbol{v})\rangle \\
&= \langle \mathbb{II}(\boldsymbol{u}, \boldsymbol{v}), \mathbb{II}(\boldsymbol{u}, \boldsymbol{v})\rangle - \langle \mathbb{II}(\boldsymbol{u}, \boldsymbol{u}), \mathbb{II}(\boldsymbol{v}, \boldsymbol{v})\rangle \\
&\geq -\langle \mathbb{II}(\boldsymbol{u}, \boldsymbol{u}), \mathbb{II}(\boldsymbol{v}, \boldsymbol{v})\rangle,
\end{aligned} \tag{96}
$$

where we have used that $R_{\mathbb{R}^D} = 0$ and $\langle \mathbb{II}(\boldsymbol{u}, \boldsymbol{v}), \mathbb{II}(\boldsymbol{u}, \boldsymbol{v})\rangle \geq 0$. Take any $\boldsymbol{v}, \boldsymbol{v}' \in T_{\boldsymbol{s}}S$. The sectional curvature $\kappa_s(\boldsymbol{v}, \boldsymbol{v}')$ satisfies

$$\kappa_s(\boldsymbol{v}, \boldsymbol{v}') = \kappa_s(\boldsymbol{u}, \boldsymbol{u}') = \langle R_S(\boldsymbol{u}, \boldsymbol{u}')\boldsymbol{u}', \boldsymbol{u}\rangle, \tag{97}$$

for any orthonormal basis $\boldsymbol{u}, \boldsymbol{u}'$ for $\mathrm{span}(\boldsymbol{v}, \boldsymbol{v}')$. So

$$\kappa_s(\boldsymbol{v}, \boldsymbol{v}') = \langle R_S(\boldsymbol{u}, \boldsymbol{u}')\boldsymbol{u}', \boldsymbol{u}\rangle \geq -\langle \mathbb{II}(\boldsymbol{u}, \boldsymbol{u}), \mathbb{II}(\boldsymbol{u}', \boldsymbol{u}')\rangle \geq -\kappa^2, \tag{98}$$

as claimed. $\qquad\square$

**Fact 11.** *For a hyperbolic triangle with side lengths $a, b, c$ and corresponding (opposite) angles $A, B, C$, we have*

$$\cosh c = \cosh a \cosh b - \sinh a \sinh b \cos C. \tag{99}$$

## E    PROOF OF RESULT (9)

In this section, we state and prove the other part of our main claims about gradient descent:

> **Theorem 12.** *Suppose that $\boldsymbol{x} = \boldsymbol{s}_\natural + \boldsymbol{z}$, with $T^{\mathrm{max}}(\boldsymbol{z}) < 1/\kappa$. Consider the constant-stepping Riemannian gradient method, with initial point $\boldsymbol{s}^0$ satisfying $d(\boldsymbol{s}^0, \boldsymbol{s}_\natural) < 1/\kappa$, and step size $\tau = \frac{1}{64}$.*
>
> $$d\left(\boldsymbol{s}^{k+1}, \boldsymbol{s}_\natural\right) \leq (1-\varepsilon) \cdot d\left(\boldsymbol{s}^k, \boldsymbol{s}_\natural\right) + CT^{\mathrm{max}}. \tag{100}$$
>
> *Here, $C$ and $\varepsilon$ are positive numerical constants.*

Together with Theorem 6, this result shows that gradient descent rapidly converges to a neighborhood of the truth of radius $C\sigma\sqrt{d}$.

*Proof.* Let

$$\bar{\boldsymbol{s}}_t = \exp\left(-t \cdot \mathrm{grad}[f](\boldsymbol{s}^k)\right) \tag{101}$$

be a geodesic joining $\boldsymbol{s}^k$ and $\boldsymbol{s}^{k+1}$, with $\bar{\boldsymbol{s}}_0 = \boldsymbol{s}^k$ and $\bar{\boldsymbol{s}}_\tau = \boldsymbol{s}^{k+1}$. Let $f_\natural$ denote a noise-free version of the objective function, i.e.,

$$f_\natural(\boldsymbol{s}) = -\langle \boldsymbol{s}, \boldsymbol{s}_\natural\rangle, \tag{102}$$

and notice that for all $s$,

$$\mathrm{grad}[f_\natural](\boldsymbol{s}) = \mathrm{grad}[f](\boldsymbol{s}) + P_{T_{\boldsymbol{s}}S}\boldsymbol{z}. \tag{103}$$

Furthermore, following calculations in Lemma 5, on $B(\boldsymbol{s}_\natural, 1/\kappa)$, the Riemannian hessian of $f_\natural$ is bounded as

$$\|\mathrm{Hess}[f_\natural](\boldsymbol{s})\| \leq 4. \tag{104}$$

Using the relationship

$$\text{grad}[f_\natural](\bar{s}_t) = \mathcal{P}_{\bar{s}_t, \bar{s}_0} \text{grad}[f_\natural](\bar{s}_0) + \int_{r=0}^{t} \mathcal{P}_{\bar{s}_t, \bar{s}_r} \text{Hess}[f_\natural](\bar{s}_r) \mathcal{P}_{\bar{s}_r, \bar{s}_0} \text{grad}[f](\bar{s}_0) \, dr, \qquad (105)$$

where $\mathcal{P}_{\bar{s}_t, \bar{s}_0}$ to denote parallel transport along the curve $\bar{s}_t$, we obtain that

$$\left\| \text{grad}[f_\natural](\bar{s}_t) - \mathcal{P}_{\bar{s}_t, \bar{s}_0} \text{grad}[f_\natural](\bar{s}_0) \right\| \leq 4t \|\text{grad}[f](\bar{s}_0)\|_2. \qquad (106)$$

Along the curve $\bar{s}_t$, the distance to $s_\natural$ evolves as

$$
\begin{aligned}
\frac{d}{dt} d\big(\bar{s}_t, s_\natural\big) &= -\left\langle \mathcal{P}_{t,0} \text{grad}[f](\bar{s}_0), \frac{-\log_{\bar{s}_t} s_\natural}{\|\log_{\bar{s}_t} s_\natural\|_2} \right\rangle \\
&= \left\langle -\text{grad}[f](\bar{s}_t), \frac{-\log_{\bar{s}_t} s_\natural}{\|\log_{\bar{s}_t} s_\natural\|_2} \right\rangle + \left\langle \text{grad}[f](\bar{s}_t) - \mathcal{P}_{t,0} \text{grad}[f](\bar{s}_0), \frac{-\log_{\bar{s}_t} s_\natural}{\|\log_{\bar{s}_t} s_\natural\|_2} \right\rangle \\
&\leq \left\langle -\text{grad}[f](\bar{s}_t), \frac{-\log_{\bar{s}_t} s_\natural}{\|\log_{\bar{s}_t} s_\natural\|_2} \right\rangle + \left\langle \text{grad}[f_\natural](\bar{s}_t) - \mathcal{P}_{t,0} \text{grad}[f_\natural](\bar{s}_0), \frac{-\log_{\bar{s}_t} s_\natural}{\|\log_{\bar{s}_t} s_\natural\|_2} \right\rangle + 2T^{\max} \\
&\leq \left\langle -\text{grad}[f](\bar{s}_t), \frac{-\log_{\bar{s}_t} s_\natural}{\|\log_{\bar{s}_t} s_\natural\|_2} \right\rangle + 4t \|\text{grad}[f](\bar{s}_0)\| + 2T^{\max} \\
&\leq -\tfrac{1}{2} d\big(\bar{s}_t, s_\natural\big) + 4td\big(\bar{s}_0, s_\natural\big) + (3 + 4t)T^{\max} \\
&\leq -\tfrac{1}{2} d\big(\bar{s}_t, s_\natural\big) + \tfrac{1}{16} d\big(\bar{s}_0, s_\natural\big) + 4T^{\max} \qquad (107)
\end{aligned}
$$

where we have used Lemma 13. Setting $X_t = d(\bar{s}_t, s_\natural)$, we have

$$\dot{X}_t \leq -\tfrac{1}{4} X_t \qquad (108)$$

whenever $X_t \geq \frac{1}{4} X_0 + 16T^{\max}$. Hence,

$$X_\tau \leq \max\left\{ e^{-\frac{\tau}{4}} X_0, \tfrac{1}{4} X_0 + 16T^{\max} \right\}, \qquad (109)$$

and so

$$d\big(s^{k+1}, s_\natural\big) \leq \exp\big(-\tfrac{1}{256}\big) \cdot d\big(s^k, s_\natural\big) + 16T^{\max}, \qquad (110)$$

as claimed.

$\square$

### E.1 SUPPORTING LEMMAS

**Lemma 13.** *Suppose that $\Delta < 1/\kappa$. For all $s \in B(s_\natural, \Delta)$, we have*

$$\left\langle -\text{grad}[f](s), \frac{-\log_s s_\natural}{\|\log_s s_\natural\|_2} \right\rangle \leq -\tfrac{1}{2} d(s, s_\natural) + T^{\max}. \qquad (111)$$

*Proof.* Notice that

$$
\begin{aligned}
\left\langle -\text{grad}[f](s), \frac{-\log_s s_\natural}{\|\log_s s_\natural\|_2} \right\rangle &= \left\langle P_{T_s S}(s_\natural + z), \frac{-\log_s s_\natural}{\|\log_s s_\natural\|_2} \right\rangle \\
&\leq \left\langle P_{T_s S} s_\natural, \frac{-\log_s s_\natural}{\|\log_s s_\natural\|_2} \right\rangle + T^{\max}. \qquad (112)
\end{aligned}
$$

Consider a unit speed geodesic $\gamma$ joining $s_\natural$ and $s$, with $\gamma(0) = s_\natural$ and $\gamma(t) = s_\natural$. Then

$$\frac{-\log_s s_\natural}{\|\log_s s_\natural\|_2} = \dot{\gamma}(t), \qquad (113)$$

and

$$\left\langle P_{T_s S} s_\natural, \frac{-\log_s s_\natural}{\|\log_s s_\natural\|_2} \right\rangle = \langle \gamma(0), \dot\gamma(t) \rangle$$

$$= \underbrace{\langle \gamma(t), \dot\gamma(t) \rangle}_{\text{this term}=0} - \int_0^t \langle \dot\gamma(s), \dot\gamma(t) \rangle \, ds$$

$$= -t\|\dot\gamma(t)\|_2^2 - \int_0^t \int_t^s \langle \ddot\gamma(r), \dot\gamma(t) \rangle \, dr \, ds$$

$$\leq -d(s, s_\natural) + \tfrac{1}{2}\kappa d^2(s, s_\natural). \tag{114}$$

In particular, this term is bounded by $-\frac{1}{2}d(s, s_\natural)$ when $\Delta < 1/\kappa$. $\qquad\square$

**Lemma 14.** *For $s \in B(s_\natural, \Delta)$, we have*

$$\left\| \mathrm{grad}[f](s) \right\| \leq d(s, s_\natural) + T^{\max} \tag{115}$$

*Proof.* Notice that

$$\left\| \mathrm{grad}[f](s) \right\| = \left\| P_{T_s S}(s_\natural + z) \right\|$$

$$\leq \left\| P_{T_s S} s_\natural \right\| + T^{\max}$$

$$\leq \left\| P_{T_s \mathbb{S}^{D-1}} s_\natural \right\| + T^{\max}$$

$$= \sin \angle(s, s_\natural) + T^{\max}$$

$$\leq d_{\mathbb{S}^{D-1}}(s, s_\natural) + T^{\max},$$

$$\leq d_S(s, s_\natural) + T^{\max}, \tag{116}$$

as claimed. $\qquad\square$

## F  ADDITIONAL EXPERIMENTAL DETAILS

### F.1  GRAVITATIONAL WAVE GENERATION

Below we introduce some details on Gravitational Wave data generation. Synthetic gravitational waveforms are generated with the PyCBC package Nitz et al. (2023) with masses uniformly drawn from $[20, 50]$ (times solar mass $M_\odot$) and 3-dimensional spins drawn from a uniform distribution over the unit ball, at sampling rate 2048Hz. Each waveform is padded or truncated to 1 second long such that the peak is aligned at the 0.9 second location, and then normalized to have unit $\ell^2$ norm. Noise is simulated as iid Gaussian with standard deviation $\sigma = 0.1$. The signal amplitude is constant $a = 1$. The training set contains 100,000 noisy waveforms, the test set contains 10,000 noisy waveforms and pure noise each, and a separate validation set constructed iid as the test set is used to select optimal template banks for MF.

### F.2  HANDWRITTEN DIGIT RECOGNITION EXPERIMENT SETUP

The MNIST training set contains 6,131 images of the digit 3. In particular, we create a training set containing 10,000 images of randomly transformed digit 3 from the MNIST training set, and a test set containing 10,000 images each of randomly transformed digit 3 and other digits from the MNIST test set. We select a random subset of 1,000 embedded points as the quantization $\hat\Xi$ of the parameter space, and construct a $k$-d tree from it to perform efficient nearest neighbor search for kernel interpolation. Parameters of the trainable TpopT are initialized using heuristics based on the Jacobians, step sizes and smoothing levels from the unrolled optimization, similar to the previous experiment. $\xi^0$ is initialized at the center of the embedding space. We use the Adam optimizer with batch size 100 and constant learning rate $10^{-3}$.

