# OpenReview forum: "TpopT: Efficient Trainable Template Optimization on Low-Dimensional Manifolds"
_ICLR.cc/2024/Conference — Submitted to ICLR 2024_

### Official Review · Reviewer_wGVm · 2023-10-26

**Soundness:** 2 fair
**Presentation:** 3 good
**Contribution:** 3 good
**Rating:** 6
**Confidence:** 3

**Summary:**

The paper proposes a scalable framework for identification of signals modeled via manifolds that searches for a best match template via optimization, in contrast with the common matched filter approach that searches for a best match in a fixed set of templates. The proposed approach uses a kernel space embedding of the manifold data points, where the navigation of the manifold uses gradient descent and is trained via the popular unrolled optimization approach.

**Strengths:**

The proposed approach relies on a combination of kernel methods and data-centric optimization of iterative approach parameter vectors and matrices.

Theoretical results provide probabilistic accuracy guarantees that depend on properties of the manifold.

**Weaknesses:**

The presentation is not always clear. The proposed algorithm is not crisply stated.

For several common applications of manifold (e.g., delay of arrival estimation and other 1-D manifolds), there is no comparison between the proposed approach and existing parametrizations (e.g., polar, spline, etc.). The parametrization have been helpful in reducing the computational complexity and the density of samples needed during navigation. In this sense, the comparison with only matched filtering is too coarse given the extent of the literature.

Some practical considerations are addressed via "brute force", e.g., pushing for global optimality by increasing the number of initializations of the algorithm.

While Section 4 says any embedding can be used, several assumptions are made as the narrative progresses.

Since this is a data-centric method, there should be more discussion of the quality and quantity of manifold sampling needed to have acceptable performance.

The experimental section does not illuminate the performance of the embedding parametrization, e.g., what is the quality of the manifold samples obtained from TPoP vs. other methods, including the aforementioned parameterized approximations. There is also no discussion of training computational complexity or storage requirements for the experiments, including a comparison to MP or other methods. Finally, the computational comparison is given only in terms of "complexity", not running time.

The potential upside to MP implemented using FFT is not limited to the noiseless case or the single-dimensional (time series) case; a comparison including this implementation would be fair.

Minor comments:

* The connection between the Jacobians in (15) and the embedding distances in (16) should be stated more explicitly.
* Similarly, the relationship between (17) and (6) should be explicitly described.
* When discussing the computational complexity of TpopT after Figure 4, the authors should revisit the description of the parameters involved.
* There is a typo after eq. (22) "?"

**Questions:**

In (16), what is the value of i?

How is the formulation of supplement Theorems 2 and 12 (in particular eq. 2) comparable to Theorem 1 in the manuscript?

Can the authors define the logarithmic map used in supplement eq. 20?

In supplement eq. 23, what does $$\Pi$$ represent?

---

> ### Author Response · Authors · 2023-11-23
> **Response to Reviewer wGVm (1/3)**
>
> We thank the reviewer for acknowledging our combined approach of kernel methods and data-centric optimization, and for appreciating the probabilistic accuracy guarantees of our theoretical results.
>
> Regarding Weaknesses:
> > The presentation is not always clear. The proposed algorithm is not crisply stated
>
> We have summarized our proposed TpopT algorithm at the end of the section 5.
>
> > there is no comparison between the proposed approach and existing parametrizations (e.g., polar, spline, etc.)
>
> Thank you for your suggestion regarding comparisons with other parameterizations. While we appreciate the suggestion, our primary objective is to showcase the exponential gains achieved through optimization over template matching (MF), which is not exclusive of the parameterizations you mentioned. This focus allows us to highlight the efficiency-accuracy trade-off, which is the core message of our paper.
>
> The motivation behind TpopT lies in addressing challenges encountered in gravitational wave (GW) data analysis. In this context, the original landscape in GW detection can often be suboptimal, prompting the need for optimization-based approaches like TpopT to improve the efficiency and accuracy of the detection process. TpopT's ability to enhance the optimization landscape makes it particularly valuable in scenarios where the existing parameterization may not yield satisfactory results.
>
> It's important to note that TpopT is versatile and can handle cases where the original parameterization is not suitable or absent. Moreover, TpopT is not limited to specific parameterization methods and can be used in conjunction with various parametrizations, including polar and spline.
>
> >Some practical considerations are addressed via "brute force"
>
> We appreciate your observation regarding the use of multiple initializations in the theoretical scaling analysis of our approach. In our practical experiments, we have found that due to the effectiveness of our proposed smoothing and learning techniques, a single initialization suffices to achieve the desired results. This empirical finding streamlines the practical implementation of our method, making it more efficient and convenient for real-world applications.
>
> > While Section 4 says any embedding can be used, several assumptions are made as the narrative progresses
>
> We appreciate your attention to the assumptions made as the narrative progresses. In Section 4, we emphasize the importance of the chosen embedding preserving pairwise distances. Specifically, we do not claim that any arbitrary embedding can be used; rather, the embedding must have the characteristic of preserving pairwise distances. As we know, Multidimensional Scaling (MDS) is known for its ability to preserve pairwise distances. As mentioned in our paper, the classical MDS setup on Euclidean distances is equivalent to Principal Component Analysis (PCA). Therefore, we opted for PCA in our experiments due to its equivalence with MDS, which ensures the preservation of pairwise distances. We will make sure to include a reference to the equivalence of PCA and MDS in the revised manuscript to provide further clarity on our choice.
>
> We apologize for the narrative style used previously. We will also include a following detailed explanation of why embedding to preserve pairwise distances is essential in the revised manuscript.: Our objective function is defined as $f = -\left< s, x \right>$. If the distance between $\xi_1$ and $\xi_2$ in the parameter space is approximately equal to the distance between $s_1$ and $s_2$ in the signal space, we can observe that $-\left< s(\xi_1), x \right>$ is approximately equal to $-\left< s(\xi_2), x \right>$ when $\xi_1$ is close to $\xi_2$. This observation highlights the improvement in our optimization landscape when pairwise distances are preserved through embedding.
>
> > there should be more discussion of the quality and quantity of manifold sampling needed to have acceptable performance
>
> Theoretically, our method requires enough samples for each $o(1/\kappa)$ neighborhood of the manifold contains at least $O(d)$ samples. Based on this requirement, it is possible to work out a sample complexity, in terms of the volume of the manifold $\mathcal S$, its curvature, and properties of the sampling distribution (e.g., a lower bound on its density over $\mathcal S$). As with the basin of attraction, smoothing reduces the curvature $\kappa$, relaxing this requirement somewhat. We will add a brief discussion of these considerations to the final version of the paper. In figure 2 of the newly added pdf, we have also added experiments exploring the relationship between the number of training samples $N$ and the performance of the proposed method.

---

> ### Author Response · Authors · 2023-11-23
> **Response continued (2/3)**
>
> > There is also no discussion of training computational complexity or storage requirements for the experiments
>
> We appreciate your interest in the computational complexity and storage requirements for our experiments. It's important to clarify that the training computational complexity, which is a one-time cost, is typically not the primary focus in signal detection applications. The primary concern often lies in the test complexity, which affects the efficiency of real-time or near-real-time detection.
>
> As for storage requirements, we do acknowledge the importance of storage requirements. This aspect is discussed in Section 7 Discussion and Limitations, noting that it can be exponential in the signal manifold dimension, which is a common characteristic shared with matched filtering when using a pre-designed template bank. We hope this clarification addresses your concern.
>
> > the computational comparison is given only in terms of "complexity", not running time
>
> Thank you for your suggestion regarding the inclusion of running time comparisons. While we understand the importance of practical running time measurements, it's essential to note that running time can vary significantly based on factors such as code optimization and software environments. Therefore, we have focused on providing a detailed analysis of algorithmic complexity in our paper, as it offers a more stable and consistent basis for comparison. This complexity analysis allows readers to assess the inherent efficiency of our approach and make informed decisions based on the algorithmic foundations rather than specific implementations. We believe that this approach provides valuable insights into the efficiency of our method and its suitability for various practical scenarios.\\
>
> > the potential upside to MP implemented using FFT is not limited to the noiseless case or the single-dimensional (time series) case
>
> We would like to clarify that our proposed TpopT method is not limited to the noiseless case or single-dimensional time series data. We have experiments in our paper that demonstrate TpopT's performance under noise. TpopT is designed to handle noisy and multi-dimensional observations effectively, which makes it a versatile approach for a wide range of practical scenarios. In the context of gravitational wave observations, where noise is a common challenge, TpopT demonstrates its effectiveness. Moreover, we have also successfully applied TpopT to the MNIST image dataset and under noise in figure 3 of the newly added pdf, which is a multi-dimensional example, showcasing its applicability beyond single-dimensional cases. Therefore, TpopT's success in gravitational wave observations and the MNIST dataset demonstrates its ability to handle noisy, multi-dimensional signals.
>
> Regarding minor comments
>
> > The connection between the Jacobians in (15) and the embedding distances in (16) should be stated more explicitly.
>
> Equation 15 shows smoothing over gradient is equivalent to smooth over loss landscape in our problem. Through smoothed loss landscape in Equation 16, we aim to convey that applying kernel smoothing in the parameter space is nearly equivalent to kernel smoothing on the signal manifold. This near-equivalence arises from the embedding’s ability to preserve pairwise distances.
>
> > Similarly, the relationship between (17) and (6) should be explicitly described.
>
> Equation 6 outlines a general Euclidean gradient descent method. In contrast, Equation 17, which is our proposed method, also employs a gradient descent framework in Euclidean space. The key differentiator of our method lies in its innovative use of linear interpolation to determine gradients for points outside the quantization grid. Additionally, our unrolling method, as illustrated in the weight matrices W, is designed to learn and adaptively adjust the step size and gradient information, further enhancing its efficacy.
>
> >When discussing the computational complexity of TpopT after Figure 4, the authors should revisit the description of the parameters involved.
>
> We map our signal into a $d$-dimensional embedding space. The primary computational complexity during the test phase arises from the number of multiplications in $ \widetilde{\nabla   \mathbf s}^T x$ totaling $d * D$. With K iterations, this leads to a complexity of $d * D * K$. Additionally, at the end of the optimization process, we calculate the correlation between the optimized template and the observation for detection purposes, which incurs a complexity of $D$. Consequently, the overall complexity for the TpopT method amounts to $D(d*K + 1)$.
>
> > There is a typo after eq. (22) "?"
>
> Thank you for pointing out the typo after eq. (22). We apologize for any confusion. In the new version of the manuscript, we will ensure that the reference is correctly displayed.

---

> > ### Author Response · Authors · 2023-11-23
> > **Response continued (3/3)**
> >
> > Regarding Question
> > > In (16), what is the value of i?
> >
> > In our approach, we first approximate the Jacobian at discrete points in the quantized set. Then, we utilize kernel interpolation techniques to compute the Jacobian at arbitrary point. It's important to note that this kernel interpolation, whether applied to signal space or the parameter space, serves as a means to estimate values at points between the discrete samples, ensuring a more continuous and accurate representation of the Jacobian across the parameter space. Therefore, in this context, 'i' represents any point situated between the discrete samples.
> >
> > > How is the formulation of supplement Theorems 2 and 12 (in particular eq. 2) comparable to Theorem 1 in the manuscript?
> >
> > Thank you for asking about the supplement theorem formulations. Theorems 2 and 12 in the supplement are essentially re-statements of the two claims in Theorem 1 of the main paper, with slight re-formulations to provide more details and better connect to the proof. In particular, the statement of Theorem 2 also specifies the bound on $\sigma$ and the concentration probability, and Theorem 12 combined with the bound on $T^\text{max}$ (from Theorem 6) gives the original claim in Theorem 1.
> >
> > > Can the authors define the logarithmic map used in supplement eq. 20?
> >
> > The logarithmic map is the inverse function of the exponential map, and it is a fundamental concept in Riemannian geometry. We appreciate your suggestion, and we acknowledge the importance of providing clear definitions for these mappings. In the final version of the supplementary material, we will include a dedicated notation section that provides comprehensive definitions and explanations of both the logarithmic map and the exponential map to ensure clarity and facilitate a deeper understanding of our methodology.
> >
> > > In supplement eq. 23, what does $\Pi$ mean?
> >
> > $\Pi$ denotes the parallel transport of tangent vectors along the geodesic $s_t$. We will add a definition in the revised version. To enhance clarity and understanding, we will incorporate a definition of this term in the revised version of our work. Thank you for highlighting this point, and we aim to provide comprehensive explanations of key concepts in our updated materials.

---

### Official Review · Reviewer_uRtk · 2023-10-30

**Soundness:** 4 excellent
**Presentation:** 3 good
**Contribution:** 4 excellent
**Rating:** 8
**Confidence:** 4

**Summary:**

In this paper, the authors provided a proof of convergence of Riemannian gradient descent on the signal manifold, and demonstrated its superior dimension scaling compared to MF. We also proposed the trainable TpopT architecture that can handle general nonparametric families of signals.  In my view, this work represents a significant accomplishment.

**Strengths:**

The authors investigated the TpopT (TemPlate OPTimization) as an alternative scalable framework for detecting low-dimensional families of signals which maintains high interpretability, proved that it has a superior dimension scaling to covering and proposed a practical TpopT framework for nonparametric signal set.

**Weaknesses:**

Please review the comments below.

**Questions:**

1. My primary concern revolves around the assumption of $\sigma$ in Theorem 1. I understand that this assumption can simplify the conclusion, but it implies that the variance of noise should be much smaller than the initialization assumption, which is exceedingly challenging in practical scenarios. Furthermore, I have observed that the proof relies on a bounded Riemannian Hessian matrix with the constant $L$, yet the main paper does not introduce any assumptions regarding the existence of the Riemannian Hessian. I propose that the authors consider removing the assumptions related to $\sigma, \tau$, and $\epsilon$ and instead directly utilize $L$ and $\tau$ to characterize the convergence rate.

2. In the main paper, the computational complexity of all methods should be presented in a tabular format.

3. In the experimental results, it is essential to include a convergence comparison of all methods to effectively demonstrate the superiority of TpopT.

4. According to Figure 5, my understanding is that when the number of hidden layers in the MLP increases, its performance may possibly become the best. If that is the case, what would be the advantage of TpopT?

---

> ### Author Response · Authors · 2023-11-23
> **Response to Reviewer uRtk (1/2)**
>
> > My primary concern revolves around the assumption of $\sigma$ in Theorem 1
>
> Thank you for your comment on the assumption of the noise. We have two assumptions of the noise, which are $\sigma \leq c/(\kappa \sqrt{ d } )$ and $\sigma \le c / (\kappa \sqrt{D})$ where $d$ is the intrinsic dimension and $D$ is the ambient space dimension. The first assumption is mild and not rather challenging to satisfy in practical situations, since $d$ in comparison to $D$ is much smaller. Under the assumption of $\sigma \leq c/(\kappa \sqrt{ d } )$, we are able to show that gradient descent exhibits linear convergence to a $\sigma \sqrt{d }$-neighborhood of $  \mathbf s_\natural$. For the second assumption on the noise, which is stronger, we use it to guarantee that $  \mathbf s^\star$ and $  \mathbf s_\natural$ are close so that when initializing in $B(  \mathbf s_\natural, \Delta)$, the gradient points toward $  \mathbf s^\star$. With this assumption, we can show that gradient descent enjoys linear convergence for all iterations $k$.
>
> >I propose that the authors consider removing the assumptions related to $\sigma$, $\tau$ and  $\epsilon$ and instead directly utilize
>  $L$ and $\tau$ to characterize the convergence rate
>
>
> Thank you for your suggestion regarding the statement of assumption. You are right in pointing out that in the proof of the main theorem, we utilized the bounded Hessian property of the objective function $f$ to bound some of the integrands. We can notice from Lemma 5 that the high probability Hessian bound is obtained due to the assumption of the noise. The reason why we want to use $\sigma$ in the assumption is that it provides an intuitive and more direct perspective on the relationship between the noise restriction and the intrinsic dimension $d$ which is $\sigma \leq c/(\kappa \sqrt{ d } )$. If we were to use $L$, we would need it to satisfy
> \begin{equation}
>         L = \frac{d^2}{dt^2} (f\circ   \mathbf\gamma)(t) \le  1 + 2\kappa^2 \Delta^2 + \kappa \Delta + \kappa \|  \mathbf z\|.
>     \end{equation}
> as proved in Lemma 5 with high probability. Even though this would work for the proof, it would miss the intuition that when we restrict the noise level (when noise acts locally), we would have nice properties such as the gradient poining toward $  \mathbf s^\star$. The reason why we want to keep this intuition is that in the later part of our paper, we utilize it to motivate the multiscale smoothing to expand the attraction basin if the noise is too large. Thus, by presenting the restriction on noise in the assumption, it is easier for readers to connect the theorem with the other parts of the paper.
>
> Regarding the $\epsilon$ in the theorem, it is treated as a constant which is standard in convergence rate theorems, similar to the constants C and c. It is not a part of the theorem assumption. Even though we can write out the exact form of the constant, it would not provide additional meaningful information for the theorem. Thus, we choose to replace the constant with $\epsilon$.
>
> > In the main paper, the computational complexity of all methods should be presented in a tabular format
>
> Thanks for the suggestion! We will include a table in the revised manuscript to concisely compare the computational complexities of all discussed methods, facilitating a clear and direct evaluation.
>
> > In the experimental results, it is essential to include a convergence comparison of all methods to effectively demonstrate the superiority of TpopT
>
> Thank you for your suggestion regarding convergence comparison. Our primary focus with TpopT is to
> highlight its efficiency in signal detection, which we believe is best demonstrated through a balance between
> accuracy and computational complexity. In Figure 5, we compare the performance of various methods
> under identical computational constraints, illustrating TpopT’s enhanced accuracy in detection tasks. It’s
> important to note that a faster convergence rate does not necessarily indicate a method’s superiority in
> signal detection. Additionally, defining a convergence rate for Matched Filtering (MF) is not straightforward,
> making direct comparisons challenging. Our approach emphasizes practical efficiency, aligning with the
> real-world requirements of effective and accurate signal detection.

---

> ### Author Response · Authors · 2023-11-23
> **Response continued (2/2)**
>
> > According to Figure 5, my understanding is that when the number of hidden layers in the MLP increases, its performance may possibly become the best
>
> Thank you for your observation regarding the performance of Multi-Layer Perceptrons (MLPs) as depicted in Figure 5. It is indeed possible that increasing the number of hidden layers in an MLP could enhance its performance, however since we want to limit the complexity of the method, the resulting hidden layers will have to have relatively small width. In our experiments, we have found that training deep networks with small width is challenging in our tasks.
>
> Additionally, it's important to acknowledge that despite the advancements in machine learning methods, Matched Filtering remains the predominant choice in many scientific applications, such as gravitational wave astronomy and seismology. This preference is largely due to the interpretability of classical techniques. In fields where understanding and explaining the decision-making process is crucial, the 'black box' nature of many machine learning methods poses a significant barrier to their adoption. Our approach with TpopT aims to bridge this gap by offering a method that not only excels in efficiency and accuracy but also moves towards greater interpretability, aligning with the critical requirements of scientific applications.

---

### Official Review · Reviewer_Y4xN · 2023-11-03

**Soundness:** 4 excellent
**Presentation:** 3 good
**Contribution:** 2 fair
**Rating:** 5
**Confidence:** 4

**Summary:**

This paper is about signal detection. It proposes to replace (parameterized) template matching via exhaustive search over the search space by optimization. The running assumption is that the template space is a manifold (e.g. translations and rotations of a prototype, or gravitational waves generated by a model which depends on a small number of parameters).

The authors use Riemannian gradient descent on the template manifold. They prove that it converges to the "best" template (at the smallest angle) provided that it is initialized sufficiently close to the global optimum, where the value of "sufficiently close" depends on the curvature. Experiments on stylized gravitational waves and rotated and shifted MNIST suggest that the proposed method performs well in some settings.

**Strengths:**

I am not on top of the latest research in learning-based template matching, but I like this paper. It is well presented, written in a sober way, the results are clear and the application is important. It is commendable that the authors derived theoretical results and identified a parameter region where their method should outperform MF. The learning strategy via unrolling Riemannian gradient descent with kernel-smoothed gradients is elegant and well motivated.

**Weaknesses:**

On the negative side, the experiments are much too stylized, especially the MNIST one. Are there no real, complex datasets where one could test the proposed methods? (I am quite sure there must be.) At the very least one shuold eavlauate the performance on the MNIST toy example with noise, including a challenging setting with a lot of noise. It is also not completely clear (even after reading F.1) how much noise \sigma = 0.3 in the gravitational wave example actually corresponds to. When you say that the signal amplitude is constant at a=1, does it mean that it's normalized so that the inf-norm is 1? (I see that for waveforms it's the l2 normalization---are "waveforms" here "templates"?) It would be great (and necessary) to considerably improve the experiments.

Another thing that I am missing is the sliding window aspect (especially in the gravitational wave application). In streaming detection applications the signal is very long and one can take advantage of the FFT to efficiently compute the dot product with a template at many shifts (if S is generated by other groups which admit a FFT then those can be included as well). In experiments in this paper (at least that is how I interpret it) S is generated by varying some physical parameters but not the shift where the template occurs. It is not clear to me what would be faster in a real streaming application, especially when the signal manifold has dimension as low as 2.

Further, one can expect the landscape like the one in Figure 4 (right) whenever the involved signals are oscillatory. This problem is well studied for example in full waveform inversion where it's known as cycle skipping. It is often addressed by moving to some optimal transport-based loss instead of l2 (dot product). I also wonder about the suitability of PCA-style dimension reduction tactics for such oscillatory signals.

**Questions:**

In Section 4 (paragraph Embedding), are there conditions on d, n, N under which the suggested dimension reduction makes sense? Does the topology of S play a role anyhow?

One other confusing thing is about interpolation to get Jacobians (not smoothing). In (12) you introduce a kernel estimate of the Jacobian at some \xi_i which is in the dataset (it needs to be since you need s_i to compute (12)), but then later you need it for arbitrary points. Do you then obtain it by linear interpolation? Just synthesizing via PCA seems to result in a globally linear space which is not what you want.

Practicalities: what is \kappa in practice? How to know it? Can you given an estimate (and compare complexity with MF) in some cases?

---

> ### Author Response · Authors · 2023-11-23
> **Response to Reviewer Y4xN (1/2)**
>
> We thank the reviewer for your helpful feedback and suggestions. We're pleased that you appreciated the paper's presentation, clear results, and the significance of its application. Your recognition of our theoretical work and the novel learning strategy is particularly encouraging.
>
> > one shuold evaluate the performance on the MNIST toy example with noise, including a challenging setting with a lot of noise
>
> We agree that we should have tested the robustness of our method against noise in the MNIST experiment. By your suggestion, we added additional MNIST experiments in figure 3 of the newly uploaded PDF in the supplementary materials with a very low signal-to-noise ratio between $-9$ dB and $-15$ dB, which shows the robustness of our method and failure of a generic MLP under noisy conditions. We also added additional experiments involving smoothing and training samples in the gravitational waves setup.\\
>
> > what is \kappa in practice? How to know it?
>
> Thank you for raising the question of estimating curvature in practice. To address your question, we ran additional experiments on estimating the curvature by fitting a polynomial in a local neighborhood around each point on the manifold. Then we estimate the associated fundamental forms by following [1].  We found the following curvature estimates for various levels of smoothing in the gravitational waves data:
>
> | Smoothing Level | 0     | 1     | 5    | 10   | 25   | 100  |
> |-----------------|-------|-------|------|------|------|------|
> | Maximum Curvature | 11.72 | 8.78  | 2.07 | 1.04 | 0.61 | 0.12 |
>
>
> TpopT controls the effect of $\kappa$ by employing a smoothing strategy in the optimization process, and as one can see, greater smoothing leads to lesser curvature. In practice, we don't necessarily need to estimate the curvature; by tuning a multi-level smoothing scheme, we can use a course-to-fine approach to find global optima. In our extensive experiments, we find that usually a single initialization suffices.
>
> > When you say that the signal amplitude is constant at a=1, does it mean that it's normalized so that the inf-norm is 1?
>
> In this context, the term 'waveform' refers to the template we used. We applied L2 normalization to these waveforms, ensuring that the L2 norm of each waveform is 1. The constant $a$ is set to 1, indicating that we maintain a fixed signal power. By doing so, we vary the power of the noise while keeping the signal magnitude constant, allowing us to effectively assess the impact of noise variations. See Figure 4 in the newly updated PDF for an example of the normalized waveform.
>
> > It is also not completely clear (even after reading F.1) how much noise \sigma = 0.3 in the gravitational wave example actually corresponds to
>
> Figure 5 in the newly uploaded PDF demonstrates observations of gravitational waves under various noise levels. These observations are based on the formula $x = s + \sigma * z$, where $s$ represents the normalized templates/waveforms (with constant magnitude), $z$ is normal Gaussian noise, and $\sigma$ denotes the noise level. The normalization process, as clearly shown in these figures, results in a small signal magnitude, making the Signal-to-Noise Ratio (SNR) a more critical factor. The figures reveal that the SNR is approximately -13 dB at a noise level of 0.1, around -19 dB at a noise level of 0.2, and about -22.5 dB when the noise level is 0.3.
>
> >Another thing that I am missing is the sliding window aspect
>
> Thank you for bringing up the important aspect of the sliding window in streaming detection applications, particularly in the context of gravitational wave detection. You are correct in noting that in our current experiments with the TpopT framework, we generate templates by varying physical parameters, but these templates have a fixed location in the signal. This approach was deliberately chosen to showcase the significant efficiency gains our method achieves in the signal dimension $d$, which is a key contribution of our current work.
>
> We acknowledge that the naive sliding window approach is inefficient for very long signals. We are currently working on a convolutional extension to TpopT, using the Fast Fourier Transform (FFT) and max pooling in the gradient descent procedure to efficiently estimate the signal as well as its shifted location. Initial experiments on synthetic examples are very promising, and we are in the process of developing the theoretical backbone for this method.
>
> ## References
> [1] F. Cazals and M. Pouget. Estimating Differential Quantities
> Using Polynomial Fitting of Osculating Jets. https://graphics.stanford.edu/courses/cs468-03-fall/Papers/cazals_jets.pdf

---

> > ### Author Response · Authors · 2023-11-23
> > **Response Continued (2/2)**
> >
> > > This problem is well studied for example in full waveform inversion where it's known as cycle skipping. It is often addressed by moving to some optimal transport-based loss instead of l2 (dot product).
> >
> > Thank you for your insightful feedback regarding the use of the L2 norm and PCA-style dimension reduction in the context of oscillatory signals. We would like to offer additional clarification on our choice of the loss function and dimension reduction technique, particularly in light of the unique characteristics of our signals and noise model.
> >
> > The simplified version of our loss function is designed as the inner product $-\left< x,s\right>$, where $x = s_\natural + z$, and $z$ represents normal Gaussian noise. We use an L2 / correlation loss as it is equivalent to maximum likelihood estimation under Gaussian noise. Using an optimal transport based loss may help with cycle skipping, but doing so may make the connection to the statistics of the noise less transparent, in particular when setting the hypothesis test decision statistic. We prefer to maintain this transparency by reorganizing the domain of optimization using PCA/MDS to mitigate cycle skipping, which produces an embedding where points in the embedded space are close if and only if they are close in an L2 sense in the original space.
> >
> >
> > > I also wonder about the suitability of PCA-style dimension reduction tactics for such oscillatory signals.
> >
> >
> > More generally, we choose PCA because it preserves pairwise distances between data points, similar to Multi-Dimensional Scaling (MDS). We aim to improve the optimization landscape by reducing dimensionality while staying faithful to the original data. The equivalence of PCA and MDS, as discussed in Carl Henrik Ek's paper ("MDS PCA Equivalence," 2021), supports our use of PCA. It shows that PCA can effectively maintain important geometric relationships in the data, akin to MDS. We will reference Carl Henrik Ek's work in our revised manuscript to better explain our methodology.
> >
> > It is important to note that while PCA itself is a linear embedding technique, the inverse process from $\xi$ to $s$ is not linear. This non-linearity in the inverse embedding is crucial for capturing the complex characteristics of the signal manifold, especially when dealing with oscillatory signals.
> >
> > Regarding Questions
> >
> > >In Section 4 (paragraph Embedding), are there conditions on d, n, N under which the suggested dimension reduction makes sense? Does the topology of S play a role anyhow?
> >
> > First, since we use a single embedding into Euclidean space, it beneficial if the manifold is homeomorphic to (a subset of) Euclidean space, since we embed in Euclidean space. Ideally, the dimension of the Euclidean space into which the manifold is embedded should be the same as the intrinsic dimension of the manifold $\mathcal S$. The number of samples $N$ is  dictated by the volume and curvature of the manifold $\mathcal S$: to accurately estimate Jacobians, we need at least order of $O(d)$ samples at an $o(1/\kappa)$ radius of each point. We appreciate the suggestions and will comment on these dependencies in the final version.
> >
> > >In (12) you introduce a kernel estimate of the Jacobian at some \xi_i which is in the dataset (it needs to be since you need s_i to compute (12)), but then later you need it for arbitrary points. Do you then obtain it by linear interpolation? Just synthesizing via PCA seems to result in a globally linear space which is not what you want.
> >
> > Thank you for your question concerning the interpolation method we use to obtain Jacobians for arbitrary points. Your understanding is correct, and we would like to provide further clarification on this process.
> >
> > In our framework, Equation (12) indeed introduces a kernel estimate of the Jacobian at points $\xi$, which are part of the dataset. These points are crucial as they allow us to compute the Jacobian estimate for arbitrary points using the available $\xi$ values.
> >
> > For arbitrary points not directly included in the dataset, we employ linear interpolation, as outlined in Equation (13). This method allows us to approximate the Jacobian at these points effectively. The linear interpolation approach is a strategic choice, balancing computational efficiency and the accuracy of the Jacobian estimation for points outside the quantization set.

---

### Official Review · Reviewer_X8Hz · 2023-11-03

**Soundness:** 2 fair
**Presentation:** 2 fair
**Contribution:** 2 fair
**Rating:** 3
**Confidence:** 4

**Summary:**

This paper considers the template matching problem :

$$\max_{s\in S} \langle s, x\rangle$$
where $x$ is a fixed observed signal, and $S$ is a manifold. $S$ is described with samples $s_1, \dots, s_n$.
The baseline for this problem is matched filtering, which enumerates the set of samples and solves $\max_{s\in \{s_1, \dots s_n\}} \langle s, x\rangle$.
First, the authors analyze the theory of Riemannian gradient descent on $S$ to solve the problem. They show that if the algorithm is initialized close enough to the solution, we get exponential convergence.
Then, the authors turn to a practical algorithm to solve the problem when one can only access samples $s_1, \dots, s_n$ describing the manifold.
They propose to learn an embedding to a lower dimensional space and optimize over it: they build anchor points $\xi_1, \dots, \xi_n$ using a dimensionality reduction technique and then construct a function $s(\xi)$ from these points, in order to maximize $\langle s(\xi), x \rangle$.
The function is constructed by approximating its Jacobian with weighted least squares. The corresponding iterations are then unrolled in a neural network to make the whole procedure learnable.
The authors validate the method on a gravitational wave detection problem and on the mnist problem, where the goal is to detect the digit "3" from the other.

**Strengths:**

The paper is quite well written and is pleasant to read.
The problem tackled here is interesting, and the numerical results are encouraging.

**Weaknesses:**

The main weakness of the paper is that it proposes a pipeline that contains many different steps that are then unrolled. The authors do not propose an ablation study where we can clearly see the benefits of each step in the pipeline: what about a method without unrolling? what about a method that implements gradient descent without the smoothing of the jacobians? What about preconditioning? What about a method directly differentiating through the embedding map $s\to \xi$? What is the impact of the number of training samples on the performance of the unrolled method? What is the role of the hyperparameters?

The other main weakness of the paper is its theoretical analysis. The proposed method is an unrolled gradient descent over the parameters $\xi$ that aims at approximating gradient descent over a **parameterization** of the manifold (i.e., $\min_\xi \langle s(\xi), x\rangle$, where ideally $s(\xi)$ describes the whole manifold $S$). The theoretical part of the paper is about Riemannian gradient descent over $S$ itself. There are, therefore, barely any links between the method proposed by the authors in practice and that studied in theory, and the efficiency of the proposed practical method is not grounded in any theory.

**Questions:**

- I think it would be great to discuss how much of the results in thm.1. are due to the linearity of the objective function: what happens when the objective function is no longer linear?
- Page 5: what is $\phi$ ? is it the same things as $s$? it is not clear what the domain of $\phi$ is.
- In the implementation of gradient descent, one needs to compute $s(\xi)$. The authors explain in detail how they approximate the Jacobian of this map, but how is $s(\xi)$ itself approximated for $\xi \notin \{\xi_1, \dots, \xi_n\}$?
- The authors choose a compactly supported kernel to reduce computations, but it still requires to compute pairwise distances : how much computations does it really gain?
- Eq.17 is an affine equation between $\xi$ and $x$. Why would we need to learn all the matrices $W(\xi_i, k)$ when we can simply learn the full linear operator, which has far fewer parameters ?


Here are some Misc. remarks:
- some citations should be in parenthesis
- The sphere $\mathbb{S}^{D-1}$ should be defined
- Trivializations (Lezcano Casado, Mario. "Trivializations for gradient-based optimization on manifolds." Advances in Neural Information Processing Systems 32 (2019).) are a good reference for the discussion around eq.6.
-Equations 7 and 8 are missing references; to the best of my knowledge these are not novel.
- A reference for multidimensional scaling, and its equivalence to PCA, is welcome.
- Fonts in figures are sometimes too small, they should be the same size as the main text's figures.
- The provided `TpopT_MNIST.ipynb` is not runnable, since the file `data_MNIST/data_dim3.pkl` is not provided. The `get_gradient` function also calls a `X_base` variable never defined. Please provide self-contained code.

---

> ### Author Response · Authors · 2023-11-23
> **Response to Reviewer X8Hz (1/4)**
>
> We thank the reviewer for a very careful review of our paper. We are glad that you found the paper pleasant to read and that you are encouraged by the numerical experiments. We also add additional experiments on MNIST digit recognition problem with noise, please see details in the newly uploaded PDF in the supplementary materials. Regarding the questions:
>
> >The main weakness of the paper is that it proposes a pipeline that contains many different steps that are then unrolled. The authors do not propose an ablation study where we can clearly see the benefits of each step in the pipeline: what about a method without unrolling?
>
> By the reviewer's suggestion, we added additional experiments showing the necessity of each component in the pipeline. Please see Figure 1,2 of the newly uploaded PDF regarding the smoothing levels and training samples. Below, we comment on specific elements of the pipeline mentioned by the reviewer:
>
> Figure 5 of the original submission compares untrained TpopT (before unrolling) with the trained TpopT (after unrolling), demonstrating that unrolling improves performance at the same complexity level. The training procedure learns a collection of Jacobian matrices, including step size and Jacobian information. And it allows us to exert precise control over how parameters are updated during the optimization process.
>
> > what about a method that implements gradient descent without the smoothing of the jacobians?
>
> Smoothing improves the landscape of navigation: From Figure 2 in our  original submission manuscript, it is evident that the landscape is significantly rougher without the application of smoothing. Additionally, we estimated the curvature of the gravitational wave manifold under various levels of smoothing in response to reviewer Y4xN:
>
> | Smoothing Level | 0     | 1     | 5    | 10   | 25   | 100  |
> |-----------------|-------|-------|------|------|------|------|
> | Maximum Curvature | 11.72 | 8.78  | 2.07 | 1.04 | 0.61 | 0.12 |
>
> The above shows that smoothing directly decreases curvature and since the optimization landscape is directly related to the curvature of the manifold, our multi-level smoothing method can avoid bad local minima during initial gradient steps by controlling the curvature $\kappa$. In figure 1 of the newly added PDF, We find that performance is at its lowest without smoothing the Jacobian, improves with moderate smoothing, and declines with excessive smoothing. Our original multi-level smoothing procedure attains the best performance.
>
> > What about preconditioning?
>
> Thank you for the suggestion around preconditioning. In response to this comment, we tried generating preconditioners by running a grid search over diagonal and symmetric matrices, however, there seem to be limited benefits and the search space is large depending on the intrinsic dimension. We also tried quasi-newton methods such as BFGS, however, we found empirically that the Hessian approximations are quite sensitive and lead to further failure points of the method.
>
> We suspect that the trainable TpopT inherently learns a preconditioner. Again, inspired by this suggestion, we experimented with adding an additional learnable preconditioning matrix to our network, and found that the performance with and without were extremely similar.

---

> ### Author Response · Authors · 2023-11-23
> **Response continued (2/4)**
>
> > What about a method directly differentiating through the embedding map $s \rightarrow \xi$?
>
> This question raises a very important point about the mapping $  \mathbf s \mapsto   \mathbf \xi$: {  while the embedding from $  \mathbf s$ to $  \mathbf \xi$ is {     linear}, and, for MDS/PCA, has a simple expression, the inverse mapping from $  \mathbf \xi$ to $  \mathbf s$ is {     nonlinear}.} To understand this issue better, consider a linear embedding $  \mathbf \xi = \varphi(  \mathbf s) =   \mathbf \Phi   \mathbf s$, with $  \mathbf \Phi \in \mathbb R^{d \times n}$. When $\varphi$ is injective (one-to-one) over the signal set $\mathcal S$, this mapping is invertible, and the inverse $\varphi^{-1}(  \mathbf \xi)$ is given by the unique intersection
> \begin{equation*}
>     \varphi^{-1}(  \mathbf \xi) = \left[   \mathbf \Phi^{\dagger}   \mathbf \xi + \mathrm{null}(  \mathbf \Phi) \right] \cap \mathcal S,
> \end{equation*}
> where $  \mathbf \Phi^{\dagger}$ is the pseudoinverse of $  \mathbf \Phi$, and $\mathrm{null}(  \mathbf \Phi)$ is its nullspace.
> This mapping is a {  nonlinear} function of $  \mathbf \xi$, because the set $\mathcal S$ is nonlinear. This is a common challenge in inverse problems with limited data: even when the measurements $  \mathbf \xi$ are a linear function of $  \mathbf s$, the reconstruction mapping (inverse) is nonlinear.
>
> In the nonparametric setting explored in Section 4 of the submitted manuscript, there is an additional challenge: not only is $\varphi^{-1}$ nonlinear, there is no explicit expression for this mapping, since there is no explicit expression for $\mathcal S$.
>
> This is why, in the nonparametric setting, we cannot simply differentiate through the mapping $  \mathbf s \mapsto   \mathbf \xi$: although the embedding $\varphi$ is available in closed form, its inverse is not. Moreover, because the inverse is nonlinear, the Jacobian of $\frac{\partial \varphi^{-1}}{\partial   \mathbf \xi}$ {  depends on $  \mathbf \xi$}. Per the reviewer's question, this is why trainable TpopT must learn {  different} matrices $  \mathbf W(   \mathbf \xi, k )$ at different points $  \mathbf \xi$ -- a single matrix $  \mathbf W$ will not work, because the inverse mapping is nonlinear.
>
> We appreciate the reviewer's questions around this crucial point, and are very happy to discuss further if there are any additional questions or comments.
>
> > What is the impact of the number of training samples on the performance of the unrolled method?
>
> We have done additional experiments on the impact of training samples on the performance of the unrolled method, the detail can be seen in Figure 2 in the newly uploaded PDF. We can see that the performance is better with more training samples.
>
> > What is the role of the hyperparameters?
>
> The main hyperparameters in our work are step size $\alpha_k$ and kernel width $\lambda_k$. In our methodology, we adopt a multi-level smoothing scheme, where we anticipate using a larger step size during the initial phase of coarse smoothing for effective global exploration and avoidance of suboptimal local minima. As the optimization process progresses and shifts towards finer smoothing, we expect a corresponding decrease in step size, facilitating precise navigation and detailed fine-tuning near the optimal solution. This approach strikes a balance between broad exploration and meticulous exploitation, thereby enhancing the overall efficiency and accuracy of our optimization process.
>
>
>
> In the TpopT framework, the choice of kernel width is instrumental in determining the accuracy of the Jacobian estimation. A broader kernel width is particularly advantageous as it encompasses a wider range of data points, leading to a more accurate and representative Jacobian. This is crucial in our framework, as a precise Jacobian is essential for effectively navigating the manifold and optimizing the template.

---

> ### Author Response · Authors · 2023-11-23
> **Response continued (3/4)**
>
> > The other main weakness of the paper is its theoretical analysis. The proposed method is an unrolled gradient descent over the parameters $\xi$ that aims at approximating gradient descent over a parameterization of the manifold (i.e. $min_\xi \left<s(\xi),x \right>$))  where ideally $s(\xi)$ describes the whole manifold ($\mathcal S$). The theoretical part of the paper is about Riemannian gradient descent over $\mathcal S$ itself. There are, therefore, barely any links between the method proposed by the authors in practice and that studied in theory, and the efficiency of the proposed practical method is not grounded in any theory.
>
> Thank you for your valuable feedback on the theoretical aspects of our paper. We understand your concern regarding the connection between our practical unrolled gradient method and the theoretical framework centered on Riemannian gradient descent.
>
> Our theory proves that gradient descent converges to an accurate estimate of $  \mathbf s_\natural$, in a neighborhood which is controlled by the curvature of the signal set $\mathcal S$. This rigorously establishes a dimension scaling advantage of optimization over matched filtering. Moreover, the theory's dependence on curvature motivates the use of multiscale smoothing, which implicitly controls the curvature of the signal manifold $\mathcal S$.
>
> Our theoretical analysis focuses on Riemannian gradident descent over the signal manifold, mostly for reasons of simplicity: the convergence of the Riemannian gradient method depends only on the geometry of the signal manifold and the statistics of the noise, and highlights the dependence of the basin of attraction on curvature. Our theory is extensible in principle to parametric gradient descent -- compared to the Riemannian theory presented in the submission, in this modified theory, the rate of convergence and size of the basin would also depend on properties of the parameterization. In particular, for smooth, bi-Lipschitz parameterizations, the analysis of the tangent noise process in Section D directly controls the maximum gradient noise at any point in the parameter space. We agree that with the reviewer that this is an important direction, and will add discussion of this point to the final version of the paper.
>
> > I think it would be great to discuss how much of the results in thm.1. are due to the linearity of the objective function: what happens when the objective function is no longer linear?
>
> Thank you for this question on the dependence of our result on the linearity of the objective function. Our convergence analysis does not make heavy use of the linearity of the objective. This analysis uses bounds on the Riemannian Hessian, which are enjoyed by the much broader class of gradient Lipschitz functions. In particular, Lemma 5 proves this property, which is then used to control the integrands in equation 23 of the appendix. Our analysis of the estimation error makes heavier use of the form of the objective as $f(  \mathbf s) = -\langle  \mathbf s,   \mathbf s_{\natural} +   \mathbf z \rangle$ -- in particular, controlling the effect of noise $  \mathbf z$ on the Riemannian gradient direction (Section D of the appendix).
>
> > Page 5: what is $\varphi$? is it the same things as $s$? it is not clear what the domain of $\varphi$ is.
>
> Apologies for any confusion in our manuscript. To clarify, $\varphi$ is the mapping function (embedding) that transforms $s$ into $\xi$. This mapping plays a crucial role in our analysis, establishing the connection between the original signal space and the transformed space. We appreciate your attention to this detail and will ensure greater clarity in our revised manuscript.
>
> > In the implementation of gradient descent, one needs to compute $s(\xi)$. The authors explain in detail how they approximate the Jacobian of this map, but how is $s(\xi)$ itself approximated for $\xi \notin \xi_1,\cdots,\xi_n$?
>
> Thank you for your insightful observation. As you correctly pointed out, we approximate the Jacobian of $s(\xi)$ as outlined in Equation 12 of our manuscript. Additionally, for values of $\xi$ that are not included in the quantization set, we employ a method of linear interpolation, which is detailed in Equation 13. This approach ensures a comprehensive and accurate representation of
> $\xi$ throughout our analysis. We appreciate your attention to these details and hope this clarification enhances your understanding of our methodology.

---

> ### Author Response · Authors · 2023-11-23
> **Response continued (4/4)**
>
> > The authors choose a compactly supported kernel to reduce computations, but it still requires to compute pairwise distances : how much computations does it really gain?
>
> During the training phase, we employ kernel interpolation to compute the Jacobian at any given point. This method ensures accuracy and comprehensiveness in our model's learning phase. In contrast, for the testing phase, we utilize a nearest neighbor approach to quantize points and obtain gradients. This strategy significantly reduces the computational complexity during testing, as it eliminates the need to consider the cost associated with interpolation at this stage.
>
> > Eq.17 is an affine equation between $\xi$ and $x$. Why would we need to learn all the matrices $W(\xi_i,k)$ when we can simply learn the full linear operator, which has far fewer parameters?
>
> While the embedding $\varphi$ is linear, as described above, the inverse mapping $\varphi^{-1}$ from $  \mathbf \xi$ to $\mathbf{s}$ is inherently {  nonlinear}. Because $\varphi^{-1}$ is nonlinear, the Jacobian $\frac{\partial \varphi^{-1}}{\partial   \mathbf \xi}$ is different at different $  \mathbf \xi$. This is why we need to learn a different Jacobian $  \mathbf W(  \mathbf \xi, k)$ at each $  \mathbf \xi$, and cannot simply use a single matrix $  \mathbf W$. We will clarify this in the final version of the paper; to corroborate this intuition, we have experimented with a version of the method which uses a single linear mapping $  \mathbf W$. Because $\varphi^{-1}$ is inherently nonlinear, this variant performs worse than all methods tested, including trained TpopT, untrained TpopT and even matched filtering.
>
> Regarding Misc. remarks:
>
> We are grateful for your insightful remarks and suggestions. In response to your comments, we will ensure that our manuscript undergoes thorough revisions for improved clarity and consistency. This includes standardizing the citation style, providing a clear definition of
> $\mathbb{S}^{D-1}$, and incorporating the cited work of Mario Lezcano Casado, 'Trivializations for gradient-based optimization on manifolds,' in connection with Equation 6, along with a relevant discussion. We will also add necessary references for Equations 7 and 8, and equivalence between PCA and MDS (Ek, Carl Henrik. "MDS PCA Equivivalence." (2021)).
>
> Regarding the figures, we acknowledge your feedback and will adjust their sizes for enhanced clarity and visual appeal.
>
> We apologize for the issues encountered with the code execution. Due to the constraints of maintaining anonymity in the submission process, and the challenges associated with the large size of the datasets for gravitational wave and MNIST, we opted to upload only the code. While we understand the importance of having access to the datasets for a comprehensive review, we were uncertain about the process of uploading them via an anonymous link that adheres to the submission's anonymity requirements. To ensure compliance with these protocols, we decided to provide only the code at this stage. We appreciate your understanding and are exploring ways to make the datasets available in a manner that respects the review process.
>
> Thank you once again for your constructive feedback, which is instrumental in refining our work.

---

### Author Response · Authors · 2023-11-23

We sincerely appreciate the valuable feedback from all reviewers on our paper. We've addressed each question and have uploaded a new PDF with updated experiment results in the supplementary materials.

These include comparisons of gravitational wave (GW) detection performances under various smoothing levels, our multi-level smoothing method's effectiveness, the impact of training sample sizes on GW experiment outcomes, MNIST example performance under different noise conditions, and an illustration of noise levels in GW signals.

Should there be any further questions or concerns, please feel free to let us know. Thank you!

---

### Meta-Review · Area_Chair_VCt4 · 2023-12-28

**Metareview:**

The paper proposes to address the problem of detecting whether a point comes from a low-dimensional manifold or not. The reference approach for this problem considered by the authors is matched filtering, which consists in checking the magnitude, given a bank of examples $s_1, \dots, s_n$ that are supposedly in that manifold, of the dot product of these points with the new point $x$, to compute $\max_i \langle s_i, x \rangle$. The authors propose to learn instead a continuous parameterization of that manifold, as $s(\xi)$, where $\xi$ is a vector, and carry out gradient ascent on $\xi$ instead. After a theoretical section that studies the speed/performance of such gradient schemes in an idealized scheme (S.3) the authors propose in S.4 a practical implementation that stacks up a few concepts. First the data points $s_1, \dots, s_n$ are projected onto a lower dimensional manifold using, e.g., MDS (i.e. PCA), resulting in $\xi_1,\dots,\xi_n$ in-sample embeddings. The relationship between these in-sample embeddings (in the sense that they are only available on the sample points) and corresponding $s$ is extended through a two step machinery: The Jacobian $Js(\xi)$ (noted $\nabla s$ in the paper) of this hypothetical continuous map $s(\xi)$ is estimated using least-squares (using a similarity coefficient computed on the embeddings), and then smoothed again using a convex combination of jacobians (through a normalized kernel averaging technique). The authors propose in addition to this a trainable scheme, where the original estimates of jacobians ($\hat\nabla s(\xi_i)$) obtained through least-squares / kernel smoothing are initialized that way, but become *trainable* (as well as other smoothing parameters).

Reviewers have highlighted the following points:
- the paper is interesting, and fairly well written.
- the experiments are weak (i.e. synthetic / MNIST) and do not offer a compelling demonstration that this method will scale to anything more challenging than 1D signals / MNIST digit out-of-distribution detection.
- there is a disconnect between a fairly idealized Riemannian gradient descent result in Section 2 and what does look like a fairly hacky (or application specific) stacking of components in Section 4. This involves [PCA + smooth estimation of Jacobians of hypothetical map $s$ in sample + smooth extension out of sample using kernels again], which is compounded by another layer of "trainability" using unrolling. In that sense I largely agree with the comments by reviewers X8Hz, wGVm, and Y4xN that this is not only a bit hacky, but also fairly confusing. See in particular comment by reviewer Y4xN: I agree with them that Eq. (13), unlike what is stated by the authors, is still not clear on how to recover $\hat\nabla s(\xi)$ for any arbitrary $\xi$. I get it that this will mostly be different kernel evaluations on $\xi$ vs. the $\xi_i$, but this omission is representative of the lack of clarity in Section 4 overall.

Overall I feel the solution is too "crafted" to work efficiently for these fairly simple problems. As a result, and although the paper has a borderline score, I recommend rejection. At this moment, the fact that the method has many crucial moving parts that are still left for the user to decide (e.g. PCA?) and a fairly "brute force" (see comment by R. wGVm) philosophy leaves too many open questions. These issues were duly highlighted by reviewers and later discussed during the rebuttal, but still need more work. I hope this discussion will prove useful for a resubmission, either by focusing more specifically on the gravitational waves task or by tackling a more ambitious experimental setup.

Issues discussed during the rebuttal discussion:

- _the theoretical study is disconnected from the algorithm proposed in the paper. The theory developed in the paper is about Riemannian gradient descent, while the proposed method works by reparameterizing the manifold._
- _The proposed method is a pilling of many different techniques [...], which makes it even further from the theoretical section of the paper._
- _The new figures 1 + 2 in the appendix are a step towards an ablation study, but are not enough to clearly demonstrate that each part of the piling is useful._

**Justification For Why Not Higher Score:**

The paper is seriously lacking in its experiments. Presentation is very uneven, some sections feel very hacky. Messaging is blurred (a bit of theory + unrelated algorithm).

**Justification For Why Not Lower Score:**

NA

---

### Decision · Program_Chairs · 2024-01-16

Reject